# Attitude Measurement for High-Spinning Projectile with a Hollow MEMS IMU Consisting of Multiple Accelerometers and Gyros

**DOI:** 10.3390/s19081799

**Published:** 2019-04-15

**Authors:** Fuchao Liu, Zhong Su, Hui Zhao, Qing Li, Chao Li

**Affiliations:** 1School of Automation, Beijing Institute of Technology, Beijing 100081, China; sz@bistu.edu.cn (Z.S.); 3120170445@bit.edu.cn (H.Z.); 3120150395@bit.edu.cn (C.L.); 2Beijing Key Laboratory of High Dynamic Navigation Technology, Beijing Information Science & Technological University, Beijing 100101, China; liqing@bistu.edu.cn

**Keywords:** high-spinning projectile, MEMS IMU, hollow structure, roll angular rate, AUKF

## Abstract

A low cost, high precision hollow structure MEMS IMU has been developed to measure the roll angular rate of a high-spinning projectile. The hollow MEMS IMU is realized by designing the scheme of non-centroid configuration of multiple accelerometers. Two dual-axis accelerometers are respectively mounted on the pitch axis and the yaw axis away from the center of mass of the high-spinning projectile. Three single-axis gyros are mounted orthogonal to each other to measure the angular rates, respectively. The roll gyro is not only used to judge the spinning direction, but also to measure and compensate for the low rotation speed of the high-spinning projectile. In order to improve the measurement accuracy of the sensor, the sensor output error is modeled and calibrated by the least square method. By analyzing the influence of noise statistical characteristics on angular rate solution accuracy, an adaptive unscented Kalman filter (AUKF) algorithm is proposed, which has a higher estimation accuracy than UKF algorithm. The feasibility of the method is verified by numerical simulation. By using the MEMS IMU device to build a semi-physical simulation platform, the solution accuracy of the angular rate is analyzed by simulating different rotation speeds of the projectile. Finally, the flight test is carried out on the rocket projectile with the hollow MEMS IMU. The test results show that the hollow MEMS IMU is reasonable and feasible, and it can calculate the roll angular rate in real time. Therefore, the hollow MEMS IMU designed in this paper has certain engineering application value for high-spinning projectiles.

## 1. Introduction

A conventional IMU is composed of three accelerometers and three gyroscopes mounted in a strap-down configuration. Accelerometers are sensors that measure specific force and gyros are sensors that measure the angular rate of rotation. MEMS IMU has wide application to precision-guided weapons [1,2], ship and aircraft navigation [3,4,5], human body orientation estimation [6,7,8,9], smart phones [10], satellite positioning systems [11,12], and so forth. Currently, there is a higher demand for inertial measurement devices in various fields, requiring higher measurement accuracy. For the measurement of motion parameters of high-spinning projectiles, the traditional measurement method and the sensors can’t meet its technical requirements. Because the gyro’s range reaches saturation in high dynamic motion, its application is limited. In recent years, in order to improve the accuracy of inertial measurement, researchers have adopted the design of inertial arrays and redundant accelerometer configurations.

The literature [13,14,15,16] presents the current research status and related application problems related to inertial arrays. Madgwick [17] used a redundant array of triple-axis accelerometers to measure the linear and angular motion of the rigid body. Seyed Moosavi [18] used the redundant array of an accelerometer to measure the motion of drilling, multi-sensors fault detection and isolation. The spatial distribution of the accelerometer was also discussed, which replaced traditional orthogonal installation. Jafari [19] analyzed the performance of redundant IMUs and their various sensor configurations. Measurement accuracy can be improved with a suitable geometric configuration. The structural vibration estimation method of Sofia airborne telescope is analyzed in reference [20] and the optimal configuration of spatially distributed accelerometers were used to accurately estimate the vibration. The motion correction of turbulence measurements on a mobile platform was realized by using a single-axis gyroscope and multiple spatially distributed accelerometers [21]. A GF-IMU consisting of five accelerometers was designed by Naseri [22] the quality characteristics of the measured parameters were provided by simulation. Onodera [23] used three dual-axis accelerometers to measure the angular velocity of the Six-DOF carrier and compensated the measurement error. The GF-IMU system was carefully evaluated. It was proved by Tan [24] that any six accelerometers configuration can be converted to the cube-type IMU with the same computational simplicity. Yang [25] designed an inertial navigation system composed of seven accelerometers and discussed its optimal configuration. After that, a GF-IMU with a nine accelerometers configuration scheme is presented in references [26,27]. With nine accelerometers, it was possible to directly estimate angular motion from spatially distributed accelerometers, thereby avoiding the extra integration step. Based on a GF-IMU composed of 12 accelerometers, an EKF filter was designed for angular motion estimation and the unbiased angular motion estimation can be obtained by using dynamic model [28,29]. Park [30] studied a GF-IMU distributed by twelve accelerometers and proposed an EKF program to estimate the direction and amplitude of angular rate through the angular acceleration term and square term. For GF-IMUs with different configurations, scholars at home and abroad have conducted a lot of research on angular motion estimation methods [31,32,33,34,35,36,37]. In addition, the installation error compensation method of GF-IMU has also been studied [38,39,40,41,42].

Based on the above research results, it can be seen that there is a relatively low efficiency for using three-channel angular calculation model and low reliability when using too many sensors, which would weaken its feasibility for application. Therefore, this paper improved the traditional IMU configuration by adjusting the non-centroid orthogonal installation position of the accelerometer to measure the angular rate of the projectile. Under the condition of gyro-assisted measurement, it not only avoids the coupling of three-channel angular motion but also improves the measurement accuracy of the angular rate. This paper is organized as follows: In Section 2, the hollow structure MEMS IMU for measuring angular rate is introduced, including the system composition and sensor installation position; In Section 3, the model and calibration of the MEMS IMU output error is introduced; The acceleration output equation and the method of solving the roll angular rate under the gyro-assisted condition are given in Section 4; An estimation method of roll angular rate based on adaptive unscented Kalman filter (AUKF) is proposed in Section 5; Simulation results are presented in Section 6 and the flight test with the semi-physical simulation platform are carried out for algorithm verification and real performance evaluation; and some concluding remarks are noted in Section 7.

## 2. The MEMS IMU Configuration Design

According to the motion characteristics and application background of the high-spinning projectile, the designed MEME-IMU needs to meet the following requirements: (1) the overall system requires a small size, light weight, low power consumption and be low cost; (2) the ability to adapt to high g value acceleration shocks; (3) the ability to adapt to high speed spinning characteristics; (4) strong anti-interference and high reliability. Due to large size, poor reliability and high cost, the platform type inertial navigation device cannot be applied to the inertial measurement of the high-spinning projectile and the gyro cannot resist the high overload shock. Considering the high-spinning characteristics, the roll angular rate is measured by the accelerometers. 

Considering the slender cylindrical shape of the high-spinning projectile, this paper designs the MEMS IMU configuration in a similar shape, using the low-cost and high-precision MEMS inertial sensors to achieve the attitude measurement of projectile through reasonable sensor configuration and installation. The measurement principle is to make use of the lever arm effect of the accelerometer. The longer the lever arm distance is, the stronger the measurement signal of the accelerometer will be and the smaller the measurement error caused by noise will be. In this paper, the MEMS IMU consists of three single-axis MEMS gyroscopes and two dual-axis MEMS accelerometers. The install location of each sensor is shown in Figure 1.

In Figure 1, the MEMS IMU PCB board is designed as a ring-shaped hollow structure with a size of Φ 100 × 25 mm and a circular diameter of 50 mm in the middle. The X-axis gyro (GX) is installed on the PCB bottom plate, the accelerometer A1 and Y-axis gyro (GY) are installed on the Y-axis PCB, the accelerometer A2 and Z-axis gyro (GZ) are installed on the Z-axis PCB. The three gyros are installed orthogonally to each other and the Y-axis gyro is used to measure the pitch angular rate, the Z-axis gyro is used to measure the yaw angular rate of the projectile. In addition, this paper also optimizes the hardware circuit of the MEMS IMU. The block diagram of the hardware circuit structure of the system is shown in Figure 2.

The measurement signals of the gyros and accelerometers in Figure 2 are converted into digital signals by the 16-bit AD data acquisition chip (AD7689), which is filtered and sent to the 32-bit ARM main controller chip (STM32F429II). Then the related algorithm is used for digital signal processing and the processed signal is transported to Flash through the RS422 serial port. In the process of signal processing, the design of the stabilized voltage power supply chip (LT3065IDD) and the timing logic circuit can ensure that the sensor works in a stable voltage environment, which is conducive to reducing noise ripple and improving the measurement accuracy of the sensors. The main parameters of the MEMS accelerometers and gyros are given in Table 1.

The gyro (ADXRS645, Analog Devices, Inc. Norwood, MA, USA) is used to measure the angular rate of the roll axis, provide rotation direction for the angular acceleration measured by the accelerometer and directly measures the angular rate at low rotational speed. The gyro (ADXRS642, Analog Devices, Inc. Norwood, MA, USA) is used to measure pitch and yaw angular rate. The accelerometer (ADXL377, Analog Devices, Inc. Norwood, MA, USA) directly measures the specific forces of the pitch axis and yaw axis.

## 3. Sensor Error Modeling and Calibration

In the actual application of the sensors, there are measurement errors and installation errors, which will affect the measurement output. In this paper, the traditional discrete calibration method is used to compensate for the error of the MEMS gyro [43]. The error model and calibration method are not described. This chapter focuses on error modeling and the calibration of the applied MEMS accelerometer.

### 3.1. Accelerometer Error Model

The main error sources of accelerometer include deterministic errors and random errors, mainly including bias error caused by noise and temperature, scale factor error (or nonlinear error), cross-coupling error and random error caused by temperature instability, and so forth. The error model can be rewritten as:
(1)$$a=CR_{a}\cdot SF_{a2}\cdot [SF_{a1}\cdot (A-N_{a0})-N_{aT}]+\varepsilon_{a}$$

Expanding Equation (1), we can get:(2)$$\left[\begin{array}{c} a_{y}\\ a_{z} \end{array}\right]=\left[\begin{array}{cc} 1 & C_{ayz}\\ C_{azy} & 1 \end{array}\right]\cdot \left[\begin{array}{cc} K_{ay2} & 0\\ 0 & K_{az2} \end{array}\right]\cdot \{\left[\begin{array}{cc} k_{ay1} & 0\\ 0 & k_{az1} \end{array}\right]\left(\left[\begin{array}{c} A_{y}\\ A_{z} \end{array}\right]-\left[\begin{array}{c} N_{ay0}\\ N_{az0} \end{array}\right]\right)-\left[\begin{array}{c} N_{aTy}\\ N_{aTz} \end{array}\right]\}+\left[\begin{array}{c} \varepsilon_{ay}\\ \varepsilon_{ay} \end{array}\right]$$
where a represents the compensated output value of accelerometer, A is the original output of accelerometer, $SF_{a1}$ represents the scale factor matrix at the initial null calibration, $N_{a0}$ represents the calculated bias matrix, $N_{aT}$ is null matrix when temperature drifts, $SF_{a2}$ represents the scale factor matrix when correcting nonlinear errors, $CR_{a}$ represents Inter-axis cross-coupling coefficient matrix, $\varepsilon_{a}$ represents the random error.

### 3.2. Accelerometer Calibration 

In the paper, the Y-axis and z-axis accelerometers are calibrated respectively by a high-speed turntable, high-low temperature box, centrifuge and other equipment. The parameters in the error model are calibrated one by one using the least square method. Figure 3 shows the calibration scheme for the four positions of the accelerometer.

(a) Parameters determine of $SF_{a1}$ and $N_{a0}$

Collect the output data of the accelerometer under the gravitational field and determine the parameters $SF_{a1}$ and $N_{a0}$ by Equation (3).
(3)$$A_{c}=SF_{a1}\cdot (A_{r}-A_{N0})$$
where $A_{r}$ is the raw output when the direction of the accelerometer’s sensitive axis is the same as the direction of gravity, $A_{N0}$ represents the raw output when the accelerometer’s sensitive axis direction is perpendicular to the direction of gravity, $A_{c}=1\;g$, the parameter $SF_{a1}$ is determined by the data in Figure 4a. Figure 4b shows the calibrated accelerometer output under the gravitational field.

Figure 4 shows the output of the accelerometer under the gravity field. The raw output of the accelerometer is shown in Figure 4a, and the calibrated output is shown in Figure 4b. The output value of the accelerometer is uniformly calibrated to the g value output.

(b) Temperature drift compensation

Accelerometers are subject to temperature changes that causes bias drift. Place the MEMS IMU on the surface of the high and low temperature chamber, set the experimental temperature range from −40 °C to 55 °C, and collect data after one hour of low temperature insulation. By collecting the temperature change data of the accelerometer in the high and low temperature box and compensating for the bias drift of the accelerometer, the least square method is used for temperature compensation.
(4)$$A_{Tc}=A_{Tr}-(q_{1}T^{2}+q_{2}T+q_{3})$$

Using the above equation to calculate the temperature bias drift coefficient, where $A_{Tc}$ is the output of accelerometer after compensation, $A_{Tr}$ is the accelerometer output during the temperature change process, $q_{1}$, $q_{2}$, $q_{3}$ are the second-order fitting coefficients, *T* represents the temperature change value. Figure 5 shows the accelerometer output curve after temperature drift compensation.

In Figure 5, the output of the accelerometer drifts with the change of temperature and the maximum drift value is −1.8 g, which is seriously affected by the high temperature. By analyzing and calculating the experimental data, the error of the accelerometer after temperature compensation was reduced by 89%.

(c) Nonlinear error compensation

The nonlinear error of the accelerometer is compensated for according to the four-position calibration scheme proposed in this paper. While collecting data, the sensitive axis of the accelerometer should be coincident with the direction of centrifugal force, data points collected include ±1 g, ±10 g, ±30 g, ±60 g, ±90 g, ±120 g, ±150 g. The least square method is used to compensate for the nonlinear error.
(5)$$A_{SF_{2}}=E_{1}A_{c2}{}^{2}+E_{2}A_{c2}+E_{3}$$

Equation (5) gives the nonlinear error model of accelerometer, where $A_{c2}$ represents the compensated accelerometer output, $A_{SF_{2}}$ represents the accelerometer output at different g values, $E_{1}$, $E_{2}$, $E_{3}$ are the second-order fitting coefficients. Figure 6 shows the accelerometer output curves after compensation at different g values.

The output of the accelerometer at different g values is shown in Figure 6, and the results of the positive output and the reverse output compensation is given. By analyzing the experimental data, the nonlinear error after compensation is reduced from 1.3% to 0.2%.

(d) Cross-coupled error compensation

The coupling error of the accelerometer is mainly caused by the installation error and the non-perpendicularity between the adjacent sensitive axis, and the result is an accelerometer output error perpendicular to the axis. The least square method is used to compensate the coupling error.
(6)$$\{\begin{array}{c} A_{CRY}=C_{1}A_{ZSF2}{}^{2}+C_{2}A_{ZSF2}+C_{3}\\ A_{CRZ}=D_{1}A_{YSF2}{}^{2}+D_{2}A_{YSF2}+D_{3} \end{array}$$
where $A_{CRY}$ is Y-axis accelerometer coupling value, $A_{CRZ}$ is Z-axis accelerometer coupling value, $A_{YSF2}$ is the output of the Y-axis accelerometer after compensation, $A_{ZSF2}$ is the output of the Z-axis accelerometer after compensation, $C_{1},C_{2},C_{3}$ and $D_{1},D_{2},D_{3}$ are the second-order fitting coefficients. Figure 7 shows the accelerometer output curve after compensation at different g values.

Figure 7a shows the accelerometer output when the Y-axis coincides with the direction of gravity in the gravitational field. The Z-axis accelerometer output is deviated due to installation error and non-orthogonal error (theoretical output is approximately zero). The output error is compensated by the least square method. The compensation result is shown in Figure 7b. Figure 7c shows the accelerometer output when the centrifugal acceleration is ±150 g, and the compensation result is shown in Figure 7d. According to the analysis and calculation, the coupling error under the gravity field is reduced from the original 25.6% to 2.3%. When the centrifugal acceleration changes within the range of ±150 g, the coupling error is reduced from the original 4.3% to 0.1%.

According to the above method, the other accelerometers of MEMS IMU are separately compensated, the effective error compensation is beneficial to improve the angular velocity and acceleration measurement accuracy.

## 4. Angular Motion Measurement

The angular motion measurement of a projectile is mainly realized by sensors such as a gyroscope and an accelerometer in three orthogonal axes. The angular rate of the projectile can be directly measured by the gyro and the attitude angle of the projectile can be calculated after integration. By installing the accelerometer at the non-center of mass of the projectile, and then decomposing the angular rate from the specific force equation by using the lever arm effect, the distance between the installation position of the accelerometer and the center of mass will affect the accuracy of solving the angular rate.

### 4.1. Output Equation of Accelerometer

To study the principle of the angular velocity calculation by accelerometers, we should first know the output equation of the accelerometers. The accelerometer measures the inertial force corresponding to the unit mass acting on the body, which is called specific force. The relation between acceleration and specific force is expressed by the following equation:(7)a=f+g
where g represents the Earth’s gravitational acceleration vector. It is worth noting that all the acceleration and the specific force vector mentioned here are relative to the inertial coordinate frame. For any point P on the projectile, the relation between the inertial frame and the body frame is shown in Figure 8.

In the Figure 8, R and $R_{0}$ denote the position vectors representing earth’s core to point P and the projectile centroid O, respectively, and r denotes the position vector of the projectile centroid O to the point P. According to the coordinate relationship, we can write:
(8)$$R=R_{0}+r$$

Equation (8) is derived twice. We can write:
(9)$${\ddot{R}}^{i}={\ddot{R}}_{0}^{i}+{\ddot{r}}^{i}$$
where ${\ddot{R}}^{i}$ is the line acceleration of point P relative to the inertial coordinate system, ${\ddot{R}}_{0}^{i}$ is the absolute line acceleration at the center of the projectile, and ${\ddot{r}}^{i}$ is the component of the line acceleration in the inertial frame.

According to the relationship between the absolute derivative and the relative derivative of the vector, the vector r is derived in the inertial frame, we get:(10)$${\dot{r}}^{i}={\dot{r}}^{b}+\omega_{ib}^{b}\times r^{b}$$
where $\omega_{ib}^{b}$ represents the component of the angular velocity in the body frame relative to the inertial frame. Equation (10) is derived again. We have the following equation:(11)$${\ddot{r}}^{i}={\ddot{r}}^{b}+2\omega_{ib}^{b}\times {\dot{r}}^{b}+\omega_{ib}^{b}\times (\omega_{ib}^{b}\times r^{b})+{\dot{\omega }}_{ib}^{b}\times r^{b}$$
where ${\ddot{r}}^{b}$ represents the component of the line acceleration of the point P in the body frame relative to the inertial frame, $2\omega_{ib}^{b}\times {\dot{r}}^{b}$ represents Coriolis acceleration, $\omega_{ib}^{b}\times (\omega_{ib}^{b}\times r^{b})$ represents centrifugal acceleration, ${\dot{\omega }}_{ib}^{b}\times r^{b}$ represents tangential acceleration of the point P, Since the point P is relatively stationary with the projectile, it is known that ${\dot{r}}^{b}={\ddot{r}}^{b}=0$, Substituting it into Equation (11), we get:(12)$${\ddot{r}}^{i}={\ddot{r}}^{b}+2\omega_{ib}^{b}\times {\dot{r}}^{b}+\omega_{ib}^{b}\times (\omega_{ib}^{b}\times r^{b})+{\dot{\omega }}_{ib}^{b}\times r^{b}$$

Substituting Equation (12) into Equation (9), we get the line acceleration equation:(13)$${\ddot{R}}^{i}={\ddot{R}}_{0}^{i}+\omega_{ib}^{b}\times (\omega_{ib}^{b}\times r^{b})+{\dot{\omega }}_{ib}^{b}\times r^{b}$$

Substituting Equation (13) into Equation (7), we get:(14)$$f^{i}=A^{i}+\omega_{ib}^{b}\times (\omega_{ib}^{b}\times r^{b})+{\dot{\omega }}_{ib}^{b}\times r^{b}$$
where $A^{i}={\ddot{R}}_{0}^{i}-g^{i}$ represents the specific force vector at the centroid. Due to the accelerometer being installed and fixed in the projectile, the measured value is the specific force component in the body frame. The specific force vector of any point in the body frame is expressed as:(15)$$f^{b}=A^{b}+\omega_{ib}^{b}\times (\omega_{ib}^{b}\times r^{b})+{\dot{\omega }}_{ib}^{b}\times r^{b}$$

We use $\theta ={\left[\cos\theta_{x},\cos\theta_{y},\cos\theta_{z}\right]}^{T}$ to represents the sensitive orientation vector and the components $\theta_{x},\theta_{y},\theta_{z}$ denote the angles between the sensitive orientation and the three axes. So the output equation of accelerometer is given as:(16)$$f=(A^{b}+\omega_{ib}^{b}\times (\omega_{ib}^{b}\times r^{b})+{\dot{\omega }}_{ib}^{b}\times r^{b})\cdot \theta^{b}$$

In order to facilitate the calculation, Equation (16) can be simplified as follows:(17)$$f=(A+\omega \times (\omega \times r)+\dot{\omega }\times r)\cdot \theta$$

Equation (16) can be described in matrix form as follows:(18)f=MV
where $M=\left[{\left[\begin{array}{c} \cos\theta_{x}\\ \cos\theta_{y}\\ \cos\theta_{z} \end{array}\right]}^{T},{\left[\begin{array}{c} \cos\theta_{y}r_{z}-\cos\theta_{z}r_{y}\\ \cos\theta_{z}r_{x}-\cos\theta_{x}r_{z}\\ \cos\theta_{x}r_{y}-\cos\theta_{y}r_{x} \end{array}\right]}^{T},{\left[\begin{array}{c} \cos\theta_{x}r_{y}+\cos\theta_{y}r_{x}\\ \cos\theta_{x}r_{z}+\cos\theta_{z}r_{x}\\ \cos\theta_{y}r_{z}+\cos\theta_{z}r_{y} \end{array}\right]}^{T},{\left[\begin{array}{c} -\cos\theta_{y}r_{y}-\cos\theta_{z}r_{z}\\ -\cos\theta_{x}r_{x}-\cos\theta_{z}r_{z}\\ -\cos\theta_{x}r_{x}-\cos\theta_{y}r_{y} \end{array}\right]}^{T}\right]$, $V=\left[A_{x},A_{y},A_{z},{\dot{\omega }}_{x},{\dot{\omega }}_{y},{\dot{\omega }}_{z},\omega_{x}\omega_{y},\omega_{x}\omega_{z},\omega_{y}\omega_{z},\omega_{x}^{2},\omega_{y}^{2},\omega_{z}^{2}\right]$, so, the required inertia vectors can be solved according to Equation (18) for different positions and different sensitive direction of the accelerometer. From the vector V we can see that no matter what kind of configuration of the accelerometer, the angular velocity of the projectile can’t be directly solved. It is possible to obtain only one or more sets of angular rate correlation quantities, including angular acceleration, angular rate product and angular rate square term. Thus $A_{x},A_{y},A_{z}$ and ${\dot{\omega }}_{x},{\dot{\omega }}_{y},{\dot{\omega }}_{z}$ are necessary output vectors for the inertial measurement system.

Equation (18) can be rewrite in matrix form as follows:(19)$$f=\left[\begin{array}{cc} \theta^{T} & {\left(\theta \times r\right)}^{T} \end{array}\right]\left[\begin{array}{c} A\\ \dot{\omega } \end{array}\right]+\theta^{T}\Omega^{2}r$$
where $\Omega =\left[\begin{array}{ccc} 0 & -\omega_{z} & \omega_{y}\\ \omega_{z} & 0 & -\omega_{x}\\ -\omega_{y} & \omega_{x} & 0 \end{array}\right]$ is the anti-symmetric matrix of ω. Consider the use of multiple accelerometers and gyroscope-free, and assume that the installation position vector is $r_{1},r_{2}\cdot \cdot \cdot r_{n}$, sensitive direction vector is $\theta_{1},\theta_{2}\cdot \cdot \cdot \theta_{n}$, the output is $f_{1},f_{2}\cdot \cdot \cdot f_{n}$, define the matrix $J_{1}=[\begin{array}{ccc} \theta_{1} & \cdot \cdot \cdot & \theta_{n} \end{array}]$ and $J_{2}=[\begin{array}{ccc} \theta_{1}\times r_{1} & \cdot \cdot \cdot & \theta_{n} \end{array}\times r_{n}]$. According to Equation (19), we get:(20)$$f=\left[\begin{array}{c} f_{1}\\ \vdots \\ f_{n} \end{array}\right]=\left[\begin{array}{cc} J_{1}^{T} & J_{2}^{T} \end{array}\right]\left[\begin{array}{c} A\\ \dot{\omega } \end{array}\right]+\left[\begin{array}{c} \theta_{1}^{T}\Omega^{2}r_{1}\\ \vdots \\ \theta_{n}^{T}\Omega^{2}r_{n} \end{array}\right]$$

Obviously, if we want to solve the specific force A and the angular acceleration, it must satisfy the condition that the matrix is reversible by Equation (20). Therefore, it must be guaranteed that rank (J) = 6, that is, at least six accelerometers can completely solve the three axial angular velocity and linear velocity of the projectile. At this point, the specific force A and $\dot{\omega }$ can be expressed as:(21)$$\left[\begin{array}{c} A\\ \dot{\omega } \end{array}\right]=J^{-1}\left[\begin{array}{c} f_{1}\\ \vdots \\ f_{n} \end{array}\right]-J^{-1}\left[\begin{array}{c} \theta_{1}^{T}\Omega^{2}r_{1}\\ \vdots \\ \theta_{n}^{T}\Omega^{2}r_{n} \end{array}\right]$$

After the angular acceleration is obtained by Equation (21), the angular rate is obtained by integrating it. Because the output of the accelerometer contains errors, the output error is transferred to the angular acceleration, and the error is spread by integrating with the angular velocity. To increase the accuracy of the calculation, it is necessary to increase the additional angular velocity related information, such as angular rate product terms and angular rate square terms, which requires an increase of the number of accelerometers.

### 4.2. Angular Rate Calculation Based on Gyro-Assisted

In the method of angular velocity calculation based on accelerometer, a higher number of accelerometers is more demanding, resulting in high cost, low precision and complicated calculation. Therefore, the method of measuring angular velocity based on gyro-assisted multi-accelerometer is the ideal method. Due to the limitations of existing gyroscopes ranges, the existing gyros cannot measure the roll angular velocity of the projectile in a highly dynamic environment, so the roll angular velocity can be obtained by the accelerometer angular velocity measurement method, yaw and pitch angular velocity can be measured directly using the gyroscope.

According to Equation (18), the measurement system only needs to output the angular velocity information related, except for the specific force of the centroid of the projectile. Based on the above analysis, the inertial measurement system obtains at least four inertia quantities in the vector V to calculate the three-axis acceleration and angular rate completely.

Assume that the measurement system outputs are $A_{x},A_{y},A_{z}$ and ${\dot{\omega }}_{x}$. Equation (18) can be rewritten in matrix form as follows:(22)$$f=\left[\begin{array}{cc} \theta^{T} & {\left(\theta \times r\right)}^{T}e_{1} \end{array}\right]\left[\begin{array}{c} A\\ {\dot{\omega }}_{x} \end{array}\right]+{\left(\theta \times r\right)}^{T}\left[\begin{array}{cc} e_{2} & e_{3} \end{array}\right]\left[\begin{array}{c} {\dot{\omega }}_{y}\\ {\dot{\omega }}_{z} \end{array}\right]+\theta^{T}\Omega^{2}r$$
where $e_{1}={\left[\begin{array}{ccc} 1 & 0 & 0 \end{array}\right]}^{T},e_{2}={\left[\begin{array}{ccc} 0 & 1 & 0 \end{array}\right]}^{T},e_{3}={\left[\begin{array}{ccc} 0 & 0 & 1 \end{array}\right]}^{T}$, Ω is the anti-symmetry matrix of ω. Consider the use of multiple accelerometers, assume that the installation position vector is $r_{1},r_{2},\cdot \cdot \cdot ,r_{n}$, sensitive direction vector is $\theta_{1},\theta_{2},\cdot \cdot \cdot ,\theta_{n}$, the output is $f_{1},f_{2},\cdot \cdot \cdot ,f_{n}$, define the matrix $J_{1}=[\begin{array}{ccc} \theta_{1} & \cdot \cdot \cdot & \theta_{n} \end{array}]$ and $J_{2}=[\begin{array}{ccc} \theta_{1}\times r_{n} & \cdot \cdot \cdot & \theta_{n} \end{array}\times r_{n}]$. According to Equation (22), we get:(23)$$f=\left[\begin{array}{c} f_{1}\\ \vdots \\ f_{n} \end{array}\right]=\left[\begin{array}{cc} J_{1}^{T} & J_{2}^{T}e_{1} \end{array}\right]\left[\begin{array}{c} A\\ {\dot{\omega }}_{x} \end{array}\right]+J_{2}^{T}\left[\begin{array}{cc} e_{2} & e_{3} \end{array}\right]\left[\begin{array}{c} {\dot{\omega }}_{y}\\ {\dot{\omega }}_{z} \end{array}\right]+\left[\begin{array}{c} \theta_{1}^{T}\Omega^{2}r_{1}\\ \vdots \\ \theta_{n}^{T}\Omega^{2}r_{n} \end{array}\right]$$

The specific force A and the angular acceleration ${\dot{\omega }}_{x}$ must satisfy that the matrix $\hat{J}=\left[\begin{array}{cc} J_{1}^{T} & J_{2}^{T}e_{1} \end{array}\right]$ is reversible. The specific force A and $\dot{\omega }$ can be expressed as:(24)$$\left[\begin{array}{c} A\\ \dot{\omega } \end{array}\right]={\hat{J}}^{-1}\left[\begin{array}{c} f_{1}\\ \vdots \\ f_{n} \end{array}\right]-{\hat{J}}^{-1}J_{2}^{T}\left[\begin{array}{cc} e_{2} & e_{3} \end{array}\right]\left[\begin{array}{c} {\dot{\omega }}_{y}\\ {\dot{\omega }}_{z} \end{array}\right]-{\hat{J}}^{-1}\left[\begin{array}{c} \theta_{1}^{T}\Omega^{2}r_{1}\\ \vdots \\ \theta_{n}^{T}\Omega^{2}r_{n} \end{array}\right]$$

Equation (24) contains angular acceleration ${\dot{\omega }}_{y}$ and ${\dot{\omega }}_{z}$, which can be removed directly due to its unobservable. In order to calculate easily, $J_{2}^{T}\left[\begin{array}{cc} e_{2} & e_{3} \end{array}\right]=0$ can be set as a constraint. The angular velocity solution accuracy can be improved by increasing the number of accelerometers.

Based on the above analysis, two dual-axis accelerometers and three single-axis gyros are used to measure the angular rate of the projectile. The configuration scheme of the sensor is shown in Figure 9.

The installation position and sensitive direction of the accelerometers in the rigid coordinate frame are shown in Figure 9. The direction indicated by the arrow is the sensitive direction of each accelerometer. The installation position of the accelerometers can be expressed as:(25)$$[r_{1},r_{2},r_{3},r_{4}]=L\left[\begin{array}{llll} 0 & 0 & 0 & 0\\ 1 & 1 & 0 & 0\\ 0 & 0 & 1 & 1 \end{array}\right]$$
where $r_{1},r_{2},r_{3},r_{4}$ respectively represents the distance between the four accelerometers and the rigid center of mass, and *L* is the designed length. The sensitive direction of each accelerometer can be expressed as:(26)$$J_{1}=[\theta_{1},\theta_{2},\theta_{3},\theta_{4}]=\left[\begin{array}{llll} 0 & 0 & 0 & 0\\ 1 & 0 & 0 & 1\\ 0 & 1 & 1 & 0 \end{array}\right]$$
where $\theta_{1},\theta_{2},\theta_{3},\theta_{4}$ respectively represents the direction of each accelerometer, the column vectors of the matrix represent the direction of the accelerometer, and the row vectors represent the X, Y and Z axes. Through Equations (25) and (26), we can get:(27)$$J_{2}=[\theta_{1}\times r_{1},\theta_{2}\times r_{2},\theta_{3}\times r_{3},\theta_{4}\times r_{4}]=\left[\begin{array}{llll} 0 & 1 & 0 & 0\\ 0 & 0 & 0 & 0\\ 0 & 0 & 0 & -1 \end{array}\right]$$

Substituting Equation (27) into Equation (23), we get:(28)$$\{\begin{array}{l} f_{1}=A_{y}-L(\omega_{x}^{2}+\omega_{z}^{2})\\ f_{2}=A_{z}+L{\dot{\omega }}_{x}+L\omega_{y}\omega_{z}\\ f_{3}=A_{z}-L(\omega_{x}^{2}+\omega_{y}^{2})\\ f_{4}=A_{y}-L{\dot{\omega }}_{x}+L\omega_{y}\omega_{z} \end{array}$$
where $f_{1},f_{2},f_{3},f_{4}$ represents the output of each accelerometer, the output of the gyroscope is $\omega_{y},\omega_{z}$, and $A_{y},A_{z}$ represents the specific force at the center of mass of the accelerometer’s sensitive axis. The angular acceleration term and square term can be expressed as:(29)$$\left\{ \begin{array}{l} {\dot{\omega }}_{x}=\frac{f_{1}-f_{4}+f_{2}-f_{3}}{2L}-\frac{\omega_{y}{}^{2}-\omega_{z}{}^{2}}{2}\\ \omega_{x}{}^{2}=\frac{f_{4}-f_{1}+f_{2}-f_{3}}{2L}-\frac{\omega_{y}{}^{2}+\omega_{z}{}^{2}}{2}-\omega_{y}\omega_{z} \end{array}\right.$$

Since $\omega_{y}$ and $\omega_{z}$ are the measured value, we only need to solve $\omega_{x}$. It can be seen from the above equation that there are two methods to calculate $\omega_{x}$—integral method and open method. The integral method is adopted to directly integrate the diagonal acceleration and the integral equation can be expressed as:(30)$$\omega_{x}(t)=\omega_{x}(t-T)+\int_{t-T}^{t}{\dot{\omega }}_{x}(\tau )\;d_{\tau }$$
where t represents the sampling time, T represents the sampling period, and ${\dot{\omega }}_{x}(\tau )$ can be determined by linear interpolation based on the angular rates at time t−T and time t. The angular rate at the current time depends on the angular rate and the angular acceleration at the previous time and the angular acceleration at the current time. Since the output error of the accelerometer is transmitted to the angular acceleration, it is finally accumulated into the angular rate of the required calculation through the superposition operation. With increasing time, the angular rate error will be larger and larger, so the integration algorithm is not conducive to work of long duration. The error does not accumulate with time, but the angular rate sign cannot be judged. Therefore, using the effective initial value measured by the X-axis gyro to judge the angular rate sign, the expression can be written as:(31)$$\omega_{x}(t)=sign(\omega_{GX})\cdot \sqrt{\omega_{x}{}^{2}(t)}$$
where $\omega_{GX}$ represents the effective initial value measured by the X-axis gyro. In this paper, the angular rate of the projectile is calculated by an open method.

## 5. Angular Motion Estimation Method

According to the output characteristics of angular velocity, the filtering algorithm can effectively improve the accuracy of angular velocity. At the same time, according to the characteristics of the output noise of the sensor, the Kalman filter algorithm is used to estimate the state. Establishing the perfect state equation and the measurement equation is the key to improving the solution. Considering the nonlinear characteristics of the system, this paper puts forward the adaptive unscented Kalman filter (AUKF) algorithm to solve the angular rate. The solution flow is shown in Figure 10.

First, the state value needs to be initialized to obtain the initial state vector $x_{0}$ and the initial error covariance matrix $P_{0}$. Calculate the sigma point and weight value $W_{i}^{m}$, and then calculate the one-step prediction status value. $X_{i,k/k-1}^{x}$ and $P_{xx,k-1}$ are the result of propagation through the equation of state. Using the UT transform, the sigma sampling point is propagated through the measurement equation to obtain $z_{i,k/k-1}$ and $P_{zz,k/k-1}$, accelerometers and gyros provide measurements in the filtering algorithm and continuously update the measurement vector $z_{k}$. After obtaining the new measurement value, the filter is updated to obtain the filter gain value $K_{k}$. From this, estimate the mean and covariance matrix of the process noise. Check if ${\hat{Q}}_{k}$ is half positive definite, if not adjust the adaptive factor $d_{k}$. Finally, the filtered state value and the covariance matrix are updated. The detailed derivation process of the algorithm is given in Section 5.2.

### 5.1. Angular Rate Calculation Model

First of all, we have to establish the state equation of the angular velocity, the angular acceleration of the projectile is considered as a random variable, which is known as the Gaussian white noise, recorded as $w_{k}={\left[w_{x,k},w_{y,k},w_{z,k}\right]}^{T}$. The sampling time is *T*. We have:(32)$$\omega_{k}=\omega_{k-1}+T{\dot{\omega }}_{k-1}$$
(33)$${\dot{\omega }}_{k}={\dot{\omega }}_{k-1}+Tw_{k-1}$$
where $\omega_{k}={\left[\omega_{x,k},\omega_{y,k},\omega_{z,k}\right]}^{T}$ is angular rate of the projectile, ${\dot{\omega }}_{k}={\left[{\dot{\omega }}_{x,k},{\dot{\omega }}_{y,k},{\dot{\omega }}_{z,k}\right]}^{T}$ is angle acceleration of the projectile. State variables $x={\left[\omega_{x},\omega_{y},\omega_{z},{\dot{\omega }}_{x},{\dot{\omega }}_{y},{\dot{\omega }}_{z},\right]}^{T}$, the state equation can be expressed as:(34)$$x_{k}=\Phi x_{k-1}+\Lambda w_{k-1}$$
where $\Phi =\left[\begin{array}{cc} I_{3\times 3} & T\cdot I_{3\times 3}\\ 0_{3\times 3} & I_{3\times 3} \end{array}\right]$, $\Lambda ={\left[\begin{array}{cc} 0_{3\times 3} & T\cdot I_{3\times 3} \end{array}\right]}^{T}$, the statistical characteristics of $w_{k}$ can be expressed as $E\left[w_{k}\right]=q_{k},E\left[w_{k}w_{j}^{T}\right]=Q_{k}\delta_{kj}$, $Q_{k}$ is symmetric semi-definite matrix, $\delta_{kj}$ is korenecker −δ function.

And then, establish the measurement equation of the system, the state variables are the angular rate and the angular acceleration of the projectile, the output of angular rate information is the angular rate or angular acceleration function, so the output of the angular rate information can be directly as measurement variables, which can be written as follows:(35)$$z_{k}=h(x_{k})+v_{k}$$
where $h(x_{k})$ is the system output information of the angular velocity, $v_{k}$ is measurement noise, and which is a linear combination of accelerometer output noise. Its statistical characteristics are $E\left[v_{k}\right]=0,E\left[v_{k}v_{j}^{T}\right]=R_{k}\delta_{kj}$, $R_{k}$ is a symmetric positive definite matrix. 

According to the configuration of MEMS IMU, the measurement equation can be expressed as:(36)$$h(x_{k})=\omega_{x,k}^{2}$$

### 5.2. Adaptive Unscented Kalman Filter Algorithm

As can be seen from the previous section, the statistical characteristics of the process noise are real-time changed, the noise estimator must be adopted in the filter algorithm. Combine the unscented Kalman filter with noise estimator to get adaptive unscented Kalman filter (AUKF). The nature of the algorithm is still UKF, and its adaptive performance in the filtering process of the process noise mean and variance of the real-time estimation. The UKF algorithm which is like the standard KF has a "predictor-corrected" algorithm structure. Respectively, we denote the dimensions of state vector, process noise vector and measurement noise vector, respectively.

The basic steps of the AUKF algorithm are as follows:

**Step 1**. Initialization
(37)$${\bar{x}}_{0}=E(x_{0})$$(38)$$P_{0}=E\left[(x_{0}-{\bar{x}}_{0}){\left(x_{0}-{\bar{x}}_{0}\right)}^{T}\right]$$

The process noise and measurement noise are augmented to the state, the state vector dimension becomes $n=n_{x}+n_{w}+n_{v}$, we get:(39)$$x^{a}={\left[x^{T},w^{T},v^{T}\right]}^{T}$$
(40)$${\bar{x}}_{0}^{a}={\left[{\bar{x}}_{0}^{T},q_{0}^{T},{\left[0\right]}_{n_{v}}\right]}^{T}$$
(41)$$P_{0}^{a}=E\left[(x_{0}^{a}-{\bar{x}}_{0}^{a}){\left(x_{0}^{a}-{\bar{x}}_{0}^{a}\right)}^{T}\right]=diag\left\{P_{0},Q_{0},R_{0}\right\}$$

**Step 2**. Calculate the Sigma sampling point and its weights

According to the proportional symmetry sampling strategy, select 2n + 1 sigma points and the corresponding first order mean weights $W_{i}^{m}$ and second-order covariance weights $W_{i}^{e}$, keep the statistical properties of the input variable x, the sampling formula can be expressed as:(42)$$X_{i}=\left[\begin{array}{ccc} \bar{x} & {\left[\bar{x}\right]}_{n}+(\sqrt{(n+\lambda )P_{xx}}) & {\left[\bar{x}\right]}_{n}-(\sqrt{(n+\lambda )P_{xx}}) \end{array}\right]$$
where ${\left[\bar{x}\right]}_{n}$ denotes a matrix which each column vector is $\bar{x}$, $\lambda =\alpha^{2}(n+k)-n$ is a scalar parameter, the weights are calculated as:(43)$$\{\begin{array}{l} W_{0}^{m}=\lambda /(n+\lambda )\\ W_{0}^{c}=\lambda /(n+\lambda )+(1-\alpha^{2}+\beta )\\ W_{i}^{m}=W_{i}^{c}=1/2(n+\lambda )\begin{array}{ccc} & & i=1,2,\ldots ,2n \end{array} \end{array}$$

Equation (43) has three parameters α,β,k, where α controls the distribution of sampling points, determines the sampling point in the surrounding spread, usually takes the value ($10^{-4}\leq \alpha \leq 1$). Constant k must ensure that the matrix positive definite, generally take the value 0 or 3 − n, β is the state distribution parameter, the general value for the Gaussian distribution is 2. According to Equations (42) and (43), we can calculate 2n + 1 sigma sampling points $X_{i,k-1}$ and corresponding weight $W_{i}^{m}$.

**Step 3**. Time update and calculate one step prediction value
(44)$$X_{i,k/k-1}^{x}=\Phi X_{i,k-1}^{x}+\Lambda X_{i,k-1}^{w}$$(45)$${\hat{x}}_{k,k-1}=\sum_{i=0}^{2n}W_{i}^{m}X_{i,k/k-1}^{x}$$(46)$$P_{xx,k-1}=\sum_{i=0}^{2n}W_{i}^{c}(X_{i,k/k-1}^{x}-{\hat{x}}_{k/k-1}){\left(X_{i,k/k-1}^{x}-{\hat{x}}_{k/k-1}\right)}^{T}$$(47)$$z_{i,k/k-1}=h(X_{i,k/k-1}^{x})+X_{i,k-1}^{v}$$(48)$${\hat{z}}_{k/k-1}=\sum_{i=0}^{2n}W_{i}^{m}z_{i,k/k-1}^{}$$

**Step 4**. Measurement equation and update equation
(49)$$P_{zz,k/k-1}=\sum_{i=0}^{2n}W_{i}^{c}(z_{i,k/k-1}^{}-{\hat{z}}_{k/k-1}){\left(z_{i,k/k-1}^{}-{\hat{z}}_{k/k-1}\right)}^{T}$$(50)$$P_{xz,k/k-1}=\sum_{i=0}^{2n}W_{i}^{c}(X_{i,k/k-1}^{x}-{\hat{x}}_{k/k-1}){\left(z_{i,k/k-1}^{}-{\hat{z}}_{k/k-1}\right)}^{T}$$

**Step 5**. Calculating the filter value and error variance matrix
(51)$$K_{k}=P_{xz,k/k-1}{\left(P_{zz,k/k-1}\right)}^{-1}$$(52)$${\tilde{z}}_{k}=z_{k}-{\hat{z}}_{k/k-1}$$(53)$$x_{k}={\hat{x}}_{k/k-1}+K_{k}{\tilde{z}}_{k}$$(54)$$P_{k}=P_{xx,k/k-1}-K_{k}P_{zz,k/k-1}K_{k}^{T}$$

**Step 6**. Calculating mean and variance of the process noise

Process noise in the state Equation (34) is estimated with noise estimator, the algorithm is as follows:(55)$$\Lambda {\hat{q}}_{k}=\frac{1}{k}\sum_{j=1}^{k}\left[{\hat{x}}_{j/k}-\Phi {\hat{x}}_{j-1/k}\right]$$
(56)$$\Lambda {\hat{Q}}_{k}\Lambda^{T}=\frac{1}{k}\sum_{j=1}^{k}\left[({\hat{x}}_{j/k}-\Phi {\hat{x}}_{j-1/k}-{\hat{q}}_{k}){\left({\hat{x}}_{j/k}-\Phi {\hat{x}}_{j-1/k}-{\hat{q}}_{k}\right)}^{T}\right]$$

The algorithm of Equation (55) and (56) needs to calculate the full smoothness of the filtered state, and the computational is too complicated to be applied in practice. Therefore, the filter estimation value and the one-step prediction value can be replaced by the full smoothing value to obtain the suboptimal noise estimator:(57)$$\Lambda {\hat{q}}_{k}=\frac{1}{k}[(k-1)\Gamma {\hat{q}}_{k-1}+{\hat{x}}_{k}-\Phi {\hat{x}}_{k-1}]$$
(58)$$\Lambda {\hat{Q}}_{k}\Lambda^{T}=\frac{1}{k}[(k-1)\Lambda {\hat{Q}}_{k-1}\Lambda^{T}+K_{k}{\tilde{z}}_{k}{\tilde{z}}_{k}^{T}K_{k}^{T}+P_{k}-\Phi P_{k-1}\Phi^{T}]$$

For the time-varying noise, the fading factor is used to fade the past data, so the mean and variance of the process noise can be estimated by the improved noise filter:(59)$$\Lambda {\hat{q}}_{k}=\frac{1}{k}[(k-1)\Gamma {\hat{q}}_{k-1}+{\hat{x}}_{k}-\Phi {\hat{x}}_{k-1}]$$
(60)$$\Lambda {\hat{Q}}_{k}\Lambda^{T}=\frac{1}{k}[(k-1)\Lambda {\hat{Q}}_{k-1}\Lambda^{T}+K_{k}{\tilde{z}}_{k}{\tilde{z}}_{k}^{T}K_{k}^{T}+P_{k}-\Phi P_{k-1}\Phi^{T}]$$
where $d_{k}=(1-b)/(1-b^{k+1})$, parameter b (0.95 < b < 0.99) denotes forgetting factor.

**Step 7**. To determine the semi-definite property of the process noise variance matrix

The improved noise estimator is an unbiased estimator, the noise statistical characteristic estimation leads to the filtering divergence, which cannot guarantee the semi-definite property of the matrix ${\hat{Q}}_{k}$. For this reason, the estimated matrix ${\hat{Q}}_{k}$ is monitored in real time. If the positive definite condition is satisfied, the value is calculated using the value of the biased noise estimator. If the value is not satisfied, the value is modified by the following equation:(61)$$\Lambda {\hat{Q}}_{k}\Lambda^{T}=\Lambda {\hat{Q}}_{k}\Lambda^{T}-d_{k}(P_{k}-\Phi P_{k-1}\Phi^{T})$$

Based on the above algorithm flow, the AUKF algorithm results from combining Sage-Husa suboptimal maximum a posteriori (MAP) noise estimator with the standard UKF.

## 6. Simulation Analysis and Flight Test

### 6.1. Simulation and Analysis

In order to verify the effectiveness of the proposed AUKF algorithm and the resolution accuracy of angular rate, the projectile spin and dynamic cone motion models are combined for simulation verification. The mathematical description of this angular motion is given as:(62)$$\omega (t)=\left[\begin{array}{c} \omega_{x}\\ \omega_{y}\\ \omega_{z} \end{array}\right]=\left[\begin{array}{c} \gamma -2\omega_{0}{\left(\sin(\alpha /2)\right)}^{2}\\ -\omega_{0}\sin(\alpha )\sin(\omega_{0}t)\\ \omega_{0}\sin(\alpha )\cos(\omega_{0}t) \end{array}\right]$$

The angular rate of spin of the projectile is set to γ=5 r/s, the angular rate of cone motion is set to $\omega_{0}=2\pi \;rad/s$ and the half cone angle is set to $\alpha =2^{\circ }$. The sampling time is chosen to be 5 ms and the total simulation time is 60 s. The reference trajectories of angular rate change are shown in Figure 11.

We seeded a white noise error of 2.7 mg (velocity random walk) in each accelerometer measurement and a white noise error of 0.02°/s (angular rate random walk) in each gyro measurement. The separation distance from the center of mass is set to L = 3.5 cm. Using the angular acceleration terms and square terms to calculate the angular rate of the projectile, and the calculate accuracy is shown in Figure 12.

In Figure 12a, the angular rate is obtained by integrating the angular acceleration term. It can be seen from the figure that the calculating error of the angular rate increases gradually with the passage of time and tends to diverge, which is mainly due to the error accumulation caused by the integral algorithm. In Figure 12b, the angular rate is obtained by calculating the square term of the angular rate. We can see that, with the passage of time, the calculating error of the angular rate does not accumulate and converges within a certain range. By comparing the two methods above, this paper uses the square term of angular rate to calculate and to use the effective initial value of the roll-axis gyro to judge the roll direction.

In Figure 13, comparison experiments were carried out to verify its effectiveness on precision improvement. The mean and covariance matrix of the process noise could be estimated in real time by a Sage-Husa suboptimal MAP noise estimator in the precondition of known measurement noise. Obviously, the UKF and AUKF algorithms proposed in this paper can correctly estimate the roll angular rate and can effectively prevent error accumulation. The AUKF algorithm has higher estimation accuracy.

The calculation accuracy of angular rate is not only related to the measurement accuracy of the accelerometer itself and the distance from the center of mass, but is also affected by the rotation speed of the projectile. In order to study the influence of projectile rotational speed on angular rate calculation accuracy, the rotational speed is set 5 r/s, 10 r/s, 15 r/s, 20 r/s, 25 r/s, and 30 r/s respectively. At the different rotational speed conditions, the calculation error of angular rate is shown in Figure 14.

It can be seen from Figure 14 that the estimation error of angular rate decreases as the rotational speed increases. When the rotational speed is less than 20 r/s, the roll angular rate estimation error tends to be stable, and there is still a small oscillation characteristic due to the influence of process noise. As the rotational speed increases, the variation of process noise leads to the increase of the estimation error of roll angle rate. As can be seen from Figure 14, the estimated error of the roll angular rate in one minute is less than 10°/s. When the rotational speed is greater than 25 r/s, the angular rate estimation error shows a divergent trend. Therefore, the MEMS IMU device in this paper is more suitable for the motion measurement of short-time working aircraft. The mean error and mean square error of angular rate are given in Table 2.

Evidently, in Table 2, as the rotation speed gets higher, the estimation errors of the angular rate get larger, which is exactly in accordance with the simulation results shown in Figure 14.

### 6.2. Semi-Physical Simulation and Analysis

The semi-physical simulation platform can evaluate the real performance of complex systems in various flight states. The main purpose of semi-physical simulation in this paper is to evaluate the calculation accuracy of the roll angular rate, and then to accurately estimate the attitude change of high-spinning projectile. The workflow diagram of the semi-physical simulation is shown in Figure 15.

The developed MEMS IMU device is mounted to the center of the three-axis turntable through a clamp, and the turntable is controlled according to a given control command. The MEMS IMU processes the acquired angular rate and angular acceleration information through a computer, according to the angular rate calculating model and the AUKF algorithm for real-time estimation of rate of roll angle.

In order to verify the accuracy of the actual roll angular rate of the MEMS IMU designed in this paper, the sensors output data are collected with the turret, which rotates at an angular rate of ±200°/s. The actual output of the gyroscope and accelerometer is shown in Figure 16.

The turntable only runs the roll axis, the pitch axis and the yaw axis are not given control commands. So, the theoretical outputs of the Y-axis and Z-axis gyro are zero. Roll axis gyro output as shown in Figure 16a, compared with the real value, the measuring error of gyro is ±0.2°/s, The output of all accelerometers is shown in Figure 16b, using the output data of the accelerometer to calculate the roll angle rate, the AUKF algorithm proposed in this paper is used to accurately estimate the roll angle rate.

Figure 17a shows the variation of the roll angular rate estimated by the accelerometer data and the AUKF algorithm. By comparing with the theoretical value, it can be seen that when the turntable roll axis is operated at an angular rate of ±200°/s, the angular rate estimation error is ±3°/s or so. The error is mainly caused by the output error of the sensor itself. It can be clearly seen from Figure 17b that the static error and the dynamic error are stable in a certain interval and do not diverge with time.

### 6.3. Flight Test

The MEMS IMU PCB designed in this paper is fixed to the hollow mechanical structure by three screws, and the internal structure of the mechanical structure is potted by a special potting process. The main purpose of the potting is to reduce the measurement error due to high overload and improve the adaptability of the projectile in special flight environment. In order to verify the practical application effect of the MEMS IMU measuring device, the device was equipped with a rocket projectile to carry out flight test verification, and also equipped with a projectile-onboard data recorder to store the flight data. The MEMS IMU and data recorder prototypes are shown in Figure 18.

During the test, the MEMS IMU device was installed at the center of mass of the projectile and fastened with screws to prevent its longitudinal sliding and axial rotation, which was beneficial in reducing the measurement error of the gyro and accelerometer. After the launch, the data recorder is recovered and the flight process data is read. Figure 19a shows the roll angle rate of the rocket projectile measured by the gyro and the roll angle rate estimated using the accelerometer data and AUKF algorithm, Figure 19b shows the error between the measured value and the estimated value.

In Figure 19, it can be seen that the overall flight time of the rocket projectile is about 12 s, the maximum rolling angular rate of the gyro measured is 1312°/s, and the error between the measured value and the estimated value is less than 6°/s. Because the rocket projectile uses two-stage engine to provide flight power, the working time of the engine in the first 2 s, the strong vibration of the projectile under the working state of the engine makes the measurement error of the sensors increase, which further increases the estimation error of the roll angular rate. It can be directly seen from Figure 19b that the roll angular rate error changes drastically in the first 2 s, and after 2 s (after the engine stopped working), the error changes smoothly and gradually converges.

Figure 20a shows the roll angle solved by the measured value of gyro and the estimated value of angular rate during the flight test. Figure 20b shows the solved error of roll angle by the two methods. The roll angle error during the whole flight test is about 7°, and the error showed a trend of divergence. There are two main reasons for the initial analysis of the error divergence. One is the error of the sensor itself, and the other is the external environment, such as wind speed disturbance. Therefore, it is necessary to study a method to suppress the drift of the roll angle error to adapt to the flight environment of the projectile and meet the test requirements.

The angular rate of the Y-axis and the Z-axis of the projectile is given in Figure 21a, and the attitude of the rocket projectile is calculated by the angular rate data. In the test, the ground frame is used as the navigation frame, and the test data is converted into the navigation frame by coordinate transformation, and then the attitude angle of the rocket projectile is calculated. From the change of the angular rate curve, the rocket projectile’s flight status can be analyzed. Under the influence of the restraint of the launching cylinder, the angular motion of rocket projectile changes sharply in the first 2 s under the thrust of the solid engine. The maximum angular velocity reaches 60°/s, and the angular velocity tends to be stable after the exit, but subject to the external environment (such as wind speed, etc.), the rocket projectile begins to sway slightly during the flight. 

It can be seen from Figure 21b that the rocket projectile launch angle is 15°, which is the initial pitch angle. The attitude angle changes drastically within 2 s after launch. The pitch angle is slightly increased by the engine’s thrust, which indicates that the rocket projectile has a significant lift effect. After the engine’s thrust is over, the pitch angle gradually becomes smaller and the rocket projectile begins to fall. Under the influence of the restraint of the launching cylinder, the initial yaw angle of the rocket changes greatly. After the launch, the yaw angle gradually converges. Therefore, it is known that the changing of pitch angle and yaw angle after the end of the engine work is relatively stable, which indicating the basic stability characteristics of the rocket projectile during the flight test. It is beneficial for algorithm verification.

## 7. Conclusions

In this paper, a hollow structure MEMS IMU device is developed, which uses the non-centroid configuration of the accelerometer to measure the roll angular rate of the projectile. In order to improve the measurement accuracy of MEMS IMU, the least square method is used to compensate for the output error of the sensors. The angular acceleration term and the square term are extracted from the specific force equation output of the accelerometer, and then the angular rate is calculated. By analyzing the influence of noise statistical characteristics on the accuracy of the angular rate calculation, an AUKF algorithm is proposed to ensure the accuracy of the angular rate calculation. Through simulation and flight test, the following conclusions are drawn:(1)The hollow structure MEMS IMU device meets the installation requirements of the internal space of the rocket projectile, which uses the lever arm effect of the accelerometer to estimate the roll angular rate and solve the problem that the roll angular rate of the projectile cannot be directly measured due to the gyro saturation.(2)The AUKF algorithm proposed in this paper is feasible and effective and has a certain suppressive effect on time-varying noise, which can improve the calculation accuracy of angular rate. However, it also has certain limitations. It takes more time and memory to calculate the statistical error mean and covariance. Therefore, it is necessary to simplify the filtering model to reduce computation time and memory.(3)The roll angular rate of the projectile is obtained by using the square terms of angular rate, and the calculation accuracy is better than the angular rate calculated by the angular acceleration terms.(4)With the increase of the speed of the projectile and the passage of time, the error of the angular rate is gradually increased. Therefore, the MEMS IMU developed in this paper is suitable for the high-spinning projectile of short-time flight.(5)The flight test verified that the feasibility of the proposed scheme. The angular velocity calculated by the accelerometer and the direct measurement of the gyroscope differ by 5°/s, and the calculated roll angle error is less than 6°. Due to the strong vibration interference during the launch of the rocket projectile and the vibration interference of the engine working process, the output error of the accelerometer becomes larger, which reduces the calculation accuracy of the angular rate.

In view of the shortcomings of the research content in this paper, future research should include the following three aspects: First, choose the accelerometer with higher accuracy, and appropriately increase the distance between the installation position of the accelerometer and the rigid center of mass to improve the measurement accuracy. Second, an initial disturbance suppression method should be studied to reduce the error of the roll angle rate. Third, to carry out the navigation algorithm research, the MEMS IMU should be equipped with higher-speed spinning projectile for flight test verification.

## Figures and Tables

**Figure 1 sensors-19-01799-f001:**
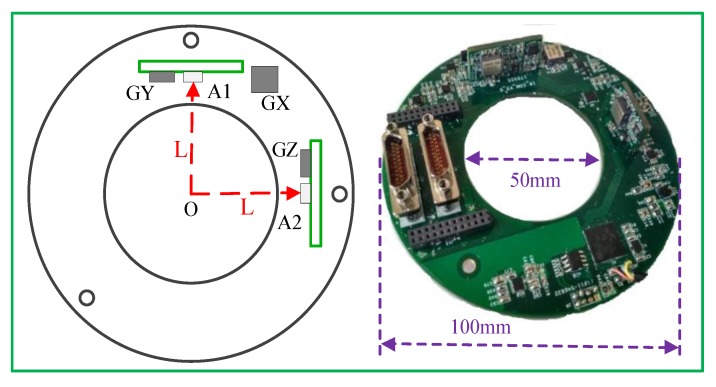
Install location of accelerometers and gyros on PCB board.

**Figure 2 sensors-19-01799-f002:**
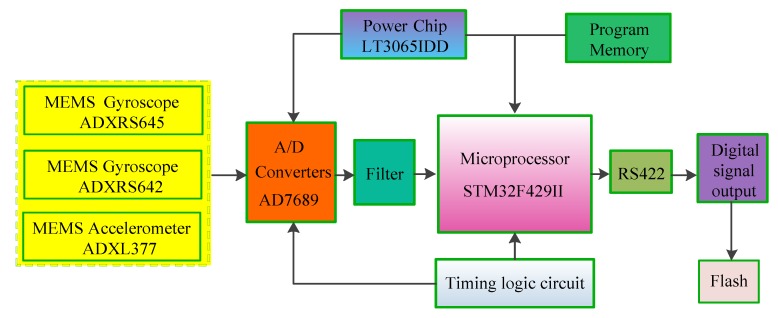
The block diagram of the hardware circuit.

**Figure 3 sensors-19-01799-f003:**
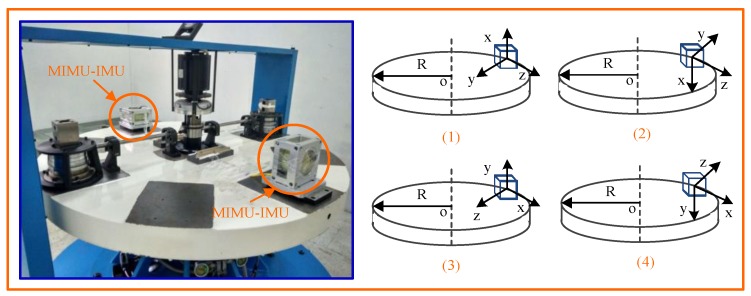
The calibration scheme based on the centrifuge.

**Figure 4 sensors-19-01799-f004:**
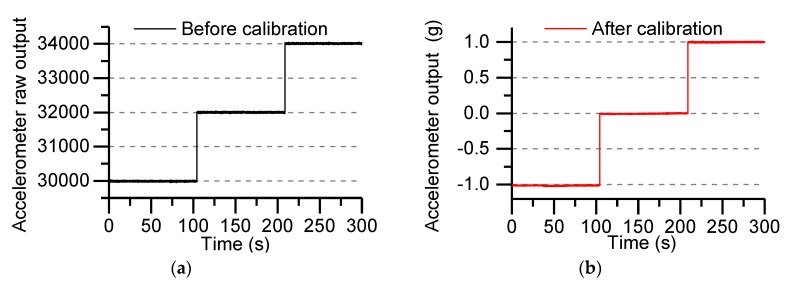
Accelerometer output under the gravitational field: (**a**) accelerometer raw output; (**b**) after calibration output of accelerometer.

**Figure 5 sensors-19-01799-f005:**
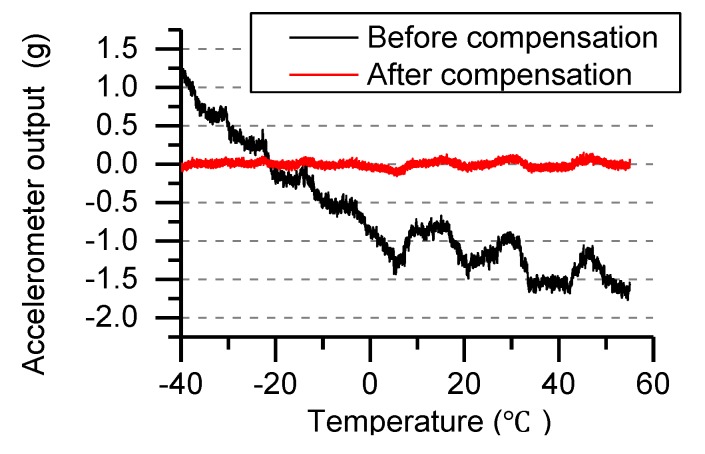
Temperature compensation.

**Figure 6 sensors-19-01799-f006:**
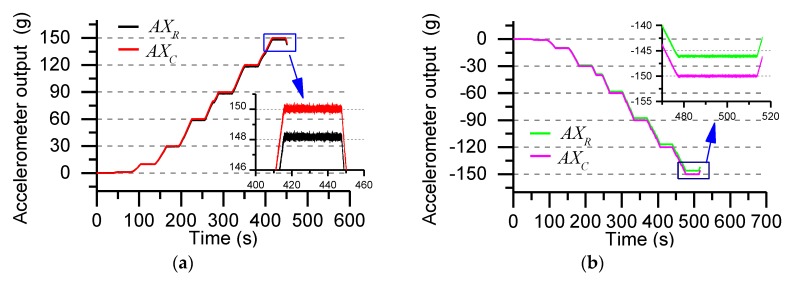
Nonlinear error compensation: (**a**) Positive output nonlinear error compensation; (**b**) Reverse output nonlinear error compensation. ($AX_{R}$ and $AX_{c}$ represent the accelerometer output before and after compensation, respectively.).

**Figure 7 sensors-19-01799-f007:**
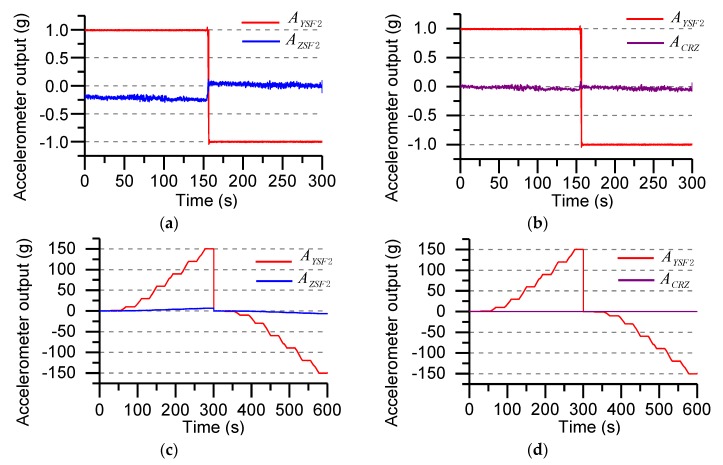
Cross-coupled error compensation of accelerometer: (**a**) Before calibration output of under the gravity field (±1 g); (**b**) After calibration output of under the gravity field (±1 g); (**c**) Before calibration output in the ±150 g acceleration range; (**d**) After calibration output in the ±150 g acceleration range.

**Figure 8 sensors-19-01799-f008:**
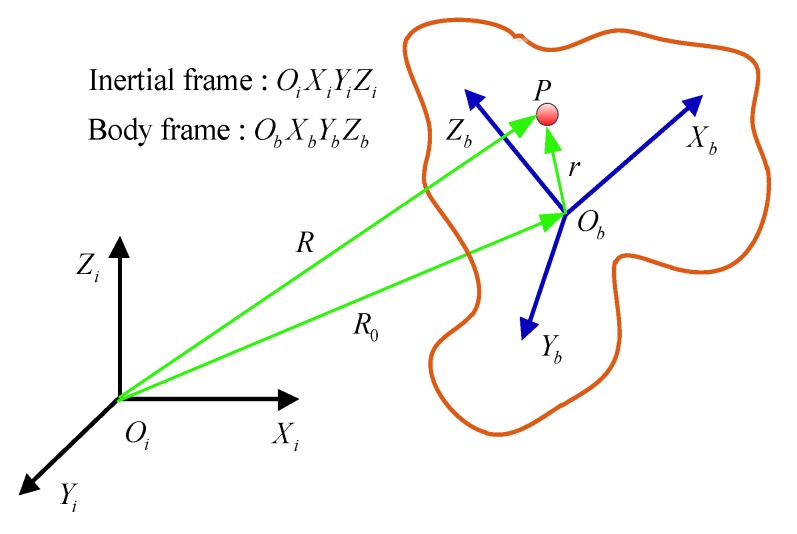
Point P in the body frame ($O_{b}X_{b}Y_{b}Z_{b}$) relative to the inertial frame ($O_{i}X_{i}Y_{i}Z_{i}$).

**Figure 9 sensors-19-01799-f009:**
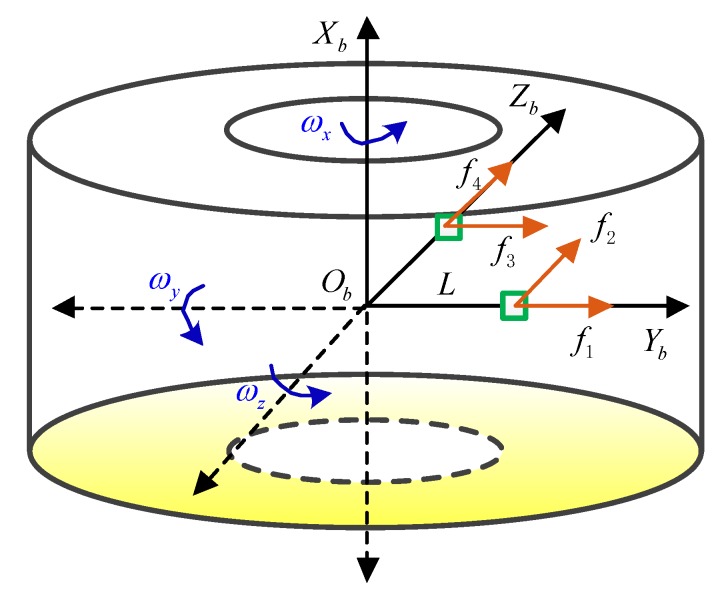
The configuration of dual-axis accelerometers in the MEMS IMU device.

**Figure 10 sensors-19-01799-f010:**
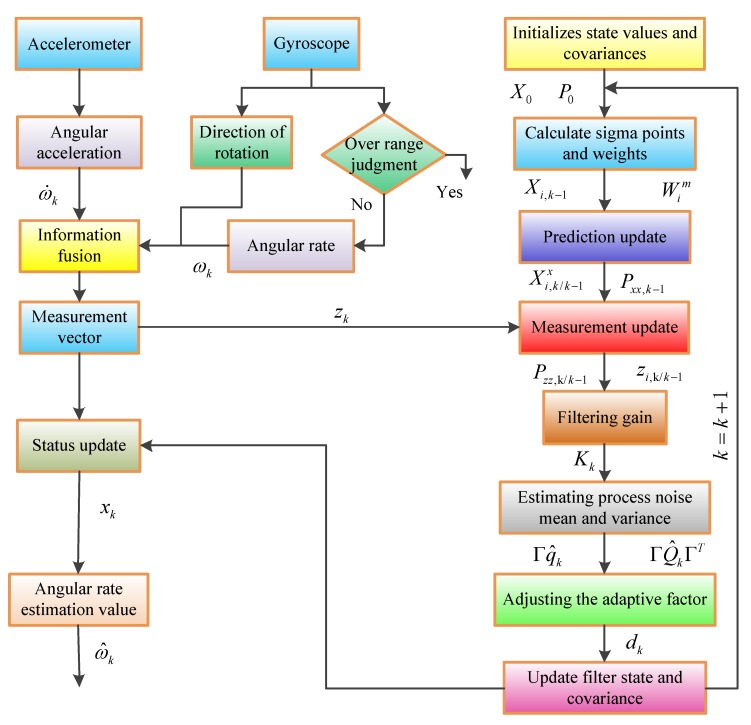
The solution flow of angular rate base on AUKF (adaptive unscented Kalman filter) algorithm.

**Figure 11 sensors-19-01799-f011:**
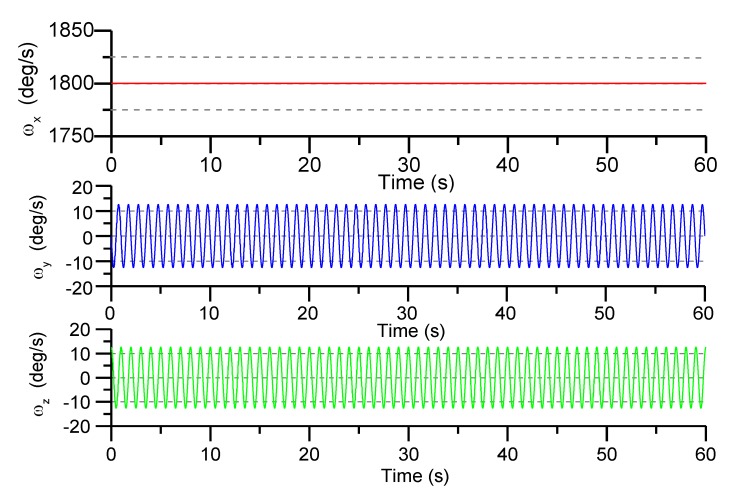
Angular rate reference curve.

**Figure 12 sensors-19-01799-f012:**
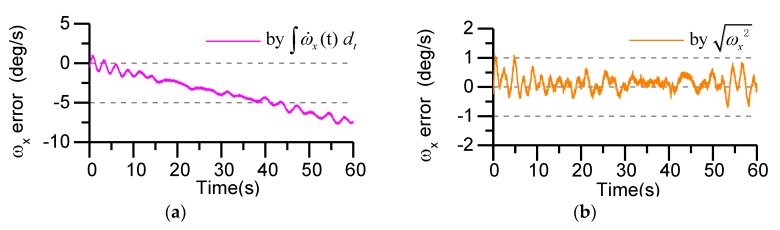
Roll angle rate calculating error: (**a**) Using the angular acceleration terms to calculate; (**b**) Using the square terms to calculate.

**Figure 13 sensors-19-01799-f013:**
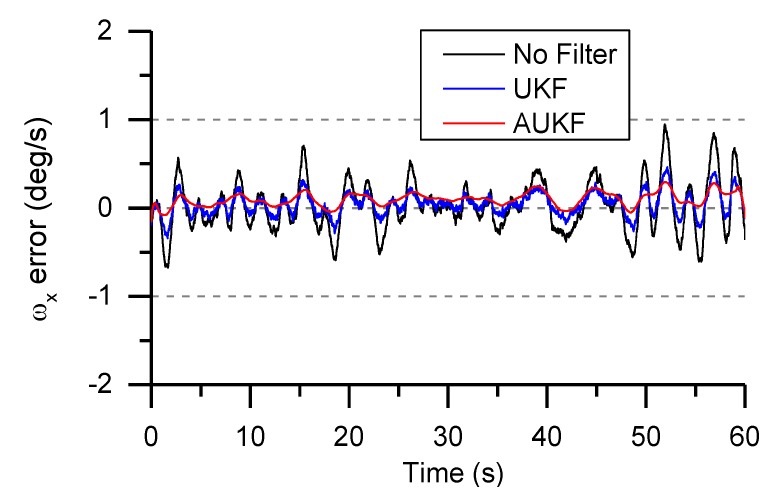
Angular rate error based on UKF and AUKF algorithm.

**Figure 14 sensors-19-01799-f014:**
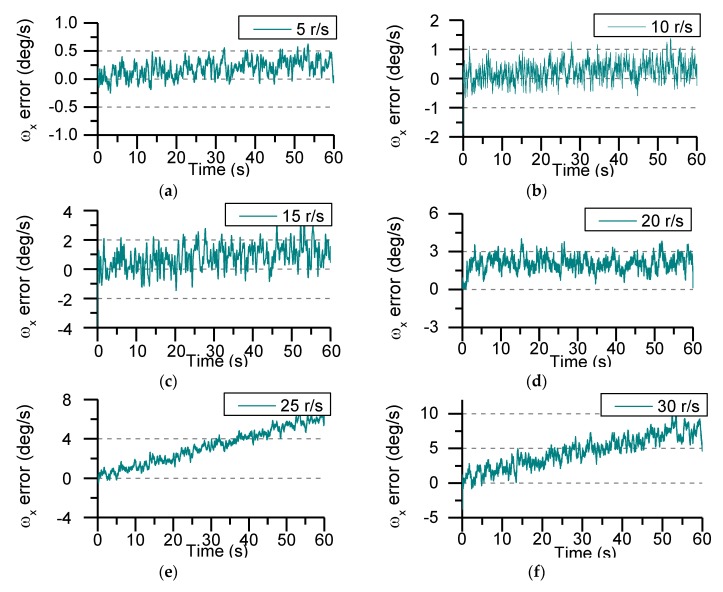
Angular rate error at different rotating speed: (**a**) The rotational speed is 5 r/s; (**b**) The rotational speed is 10 r/s; (**c**) The rotational speed is 15 r/s; (**d**) The rotational speed is 20 r/s; (**e**) The rotational speed is 25 r/s; (**f**) The rotational speed is 30 r/s.

**Figure 15 sensors-19-01799-f015:**
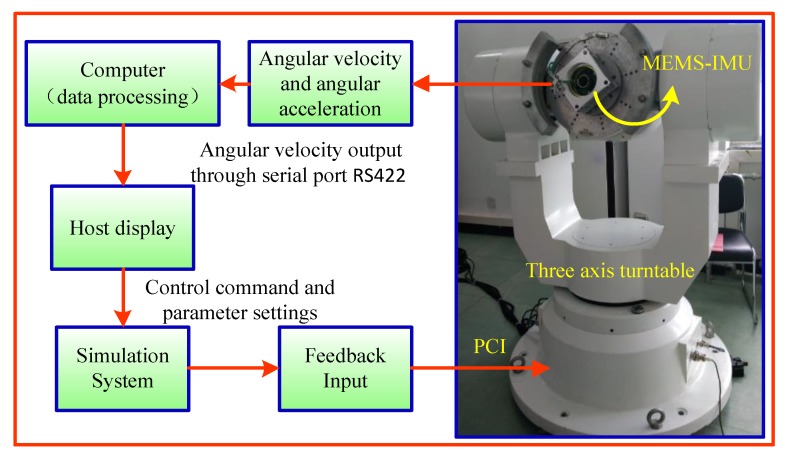
The workflow diagram of the semi-physical simulation.

**Figure 16 sensors-19-01799-f016:**
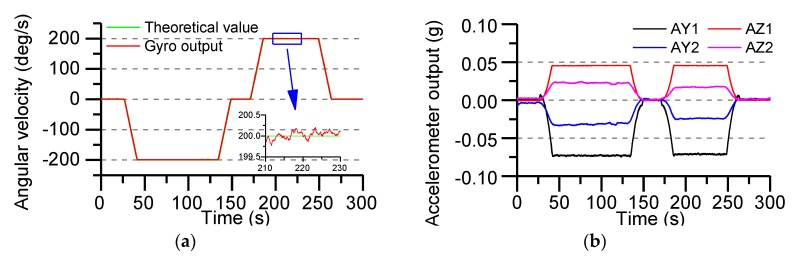
The output of sensors data (roll angle rate of the turntable is ±200°/s): (**a**) X-axis gyro output; (**b**) Accelerometers output.

**Figure 17 sensors-19-01799-f017:**
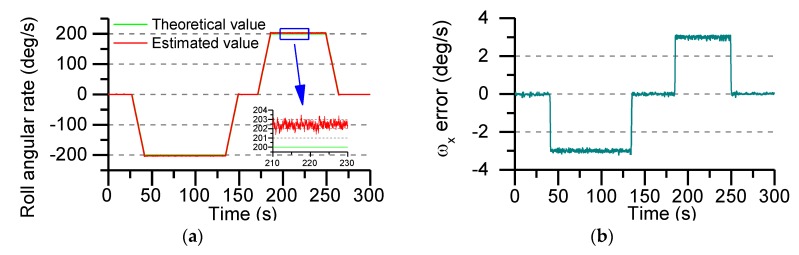
The estimated error of the roll rate using the AUKF algorithm: (**a**) Estimated value of roll angular rate; (**b**) Roll angle rate error.

**Figure 18 sensors-19-01799-f018:**
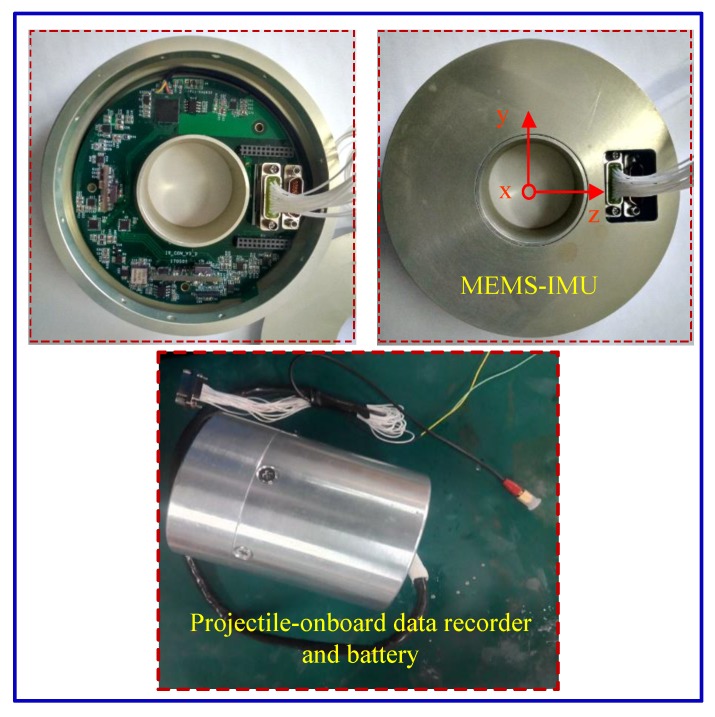
The MEMS IMU device and data recorder prototypes in the flight test.

**Figure 19 sensors-19-01799-f019:**
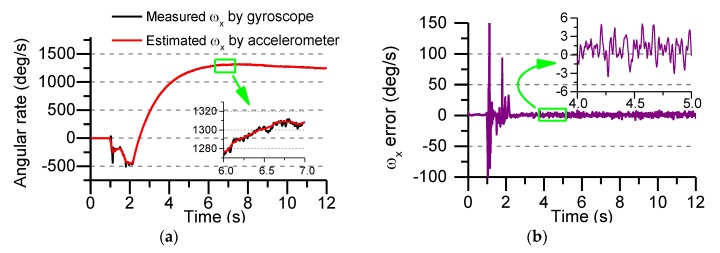
Roll angular rate and error in flight test: (**a**) The measured and estimated roll angular rate; (**b**) The error between the measured value and the estimated value.

**Figure 20 sensors-19-01799-f020:**
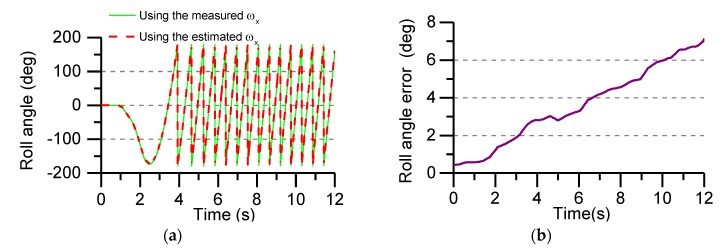
Roll angle calculated in real time and the error of roll angle in flight test: (**a**) The calculated roll angle uses the measured and estimated angular rate; (**b**) The error of roll angle.

**Figure 21 sensors-19-01799-f021:**
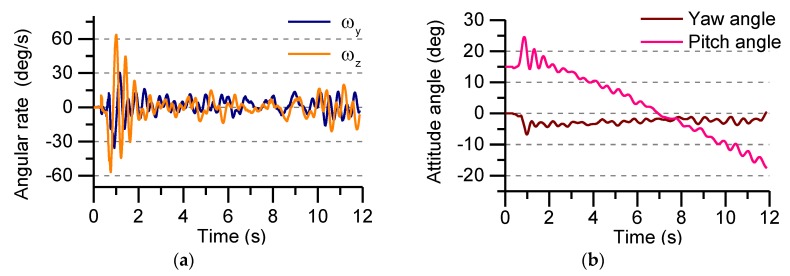
The Y-axis and z-axis angular rate and attitude angle in flight test: (**a**) The Y-axis and z-axis angular rate; (**b**) Yaw angle and Pitch angle.

**Table 1 sensors-19-01799-t001:** The main parameters of the MEMS accelerometers and gyros.

Parameter	Gyroscope (ADXRS 645)	Gyroscope (ADXRS 642)	Accelerometer (ADXL377)
Range	±2000°/s	±300°/s	±200 g
Bias Instability	100°/h	20°/h	±12 mg
Non-linearity	0.1% of FS	0.01% of FS	±0.5%
Noise Density	0.25°/s/√Hz	0.02°/s/√Hz	2.7 mg/√Hz
Operating Voltage	5 V	5 V	3 V
Bandwidth	2000 Hz	2000 Hz	1300 Hz

**Table 2 sensors-19-01799-t002:** Mean value and mean square deviation of angular rate estimated error.

Rotation Speed (r/s)	Mean Absolute Deviation (°/s)	Root-mean-square Error $\left(\sqrt{{}^{\circ}/s}\right)$
5	0.1948	5.1564 × 10^−5^
10	0.2521	2.2582 × 10^−4^
15	0.8136	1.6326 × 10^−4^
20	2.1147	7.1774 × 10^−3^
25	3.2317	7.8361 × 10^−2^
30	4.5782	1.2682 × 10^−2^

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
