# Peer review of "Attitude Measurement for High-Spinning Projectile with a Hollow MEMS IMU Consisting of Multiple Accelerometers and Gyros"

_sensors, 2019, doi:10.3390/s19081799_

Reviewer 1 Report

This manuscript presents a hollow structure MEMS-IMU device that uses a non-centroid configuration of the accelerometer to measure the roll angular rate of a projectile.In addition, this manuscript describes a least square method, which is used to compensate the output error of the MEMS-IMU device. However, this manuscript must be improved considering the following comments:

-Introduction section. Authors must include more recent references (e.g., references between 2015 and 2019).

Section of angular motion estimation method. Authors must add more discussions about the components of Figure 10.

Authors must include more discussions about the results of the Figure 14 and 21.

Authors must mention the main limitations of their hollow structure MEMS-IMU device and AUKF algorithm.

Author Response

Point 1: Introduction section. Authors must include more recent references (e.g., references between 2015 and 2019). 

Response 1: Thank you very much for your kind suggestion. Introduction section, thirteen references are added as follows:

[1]Skog,I.; Nilsson,J.O.; Handel,P.; Nehorai, A. Inertial sensor arrays, maximum likelihood, and Cramer-Rao bound,” IEEE Trans. Signal Process., vol. 64, no. 16, pp. 4218–4227, Aug. 2016.

[2] Wahlström,J.;  Skog,I.;  Händel,P. Inertial Sensor Array Processing with Motion Models. In Proceedings of 2018 21st International Conference on Information Fusion (FUSION), Cambridge, UK, 2018; pp.788–793.

[3] Nilsson,J.O.; Skog,I. Inertial sensor arrays — A literature review. in European Navigation Conf., Helsinki, Finland, May 2016.

[4]Schwaab, M.; Reginya,S. A.; Sikora, A.; Abramov,E.V. Measurement analysis of multiple MEMS sensor array. In IEEE Int. Conf. Integr. Navigation Syst., Saint Petersburg, Russia, May 2017.

[5]Seyed Moosavi,S.;Moaveni,B.;Moshiri,B.;Arvan,M. Auto-Calibration and Fault Detection and Isolation of Skewed Redundant Accelerometers in Measurement While Drilling Systems. Sensors 2018,18,702.

[6] Jafari,M. Optimal redundant sensor configuration for accuracy increasing in space inertial navigation system. Aerosp. Sci. Technol. 2015, 47, 467–472.

[7] Kaswekar,P.; Wagner, J. F. Sensor fusion based vibration estimation using inertial sensors for a complex lightweight structure. IEEE 2015 DGON Inertial Sensors and Systems Symposium (ISS), Karlsruhe, Germany, 22-23 September, 2015.

[8]Tjhai, C.; Keefe,K.O. Step-size estimation using fusion of multiple wearable inertial sensors. In IEEE Int. Conf. Indoor Positioning and Indoor Navigation, Sapporo, Japan, Sep. 2017.

[9] Liu,F.C.; Su,Z.; Li,Q.; Li,C.; Zhao.H.  Error Characteristics and Compensation Methods of MIMU with-Non-centroid Configurations. China Control Conference. IEEE, 2018:4877-4882.

[10]Trkov,M.; Chen,K.; Yi,J; Liu,T. Inertial Sensor-Based Slip Detection in Human Walking. IEEE TRANSACTIONS ON AUTOMATION SCIENCE AND ENGINEERING,2018.

[11] Ahmed, H.;  Tahir,M. Improving the accuracy of human body orientation estimation with wearable IMU sensors. IEEE Trans. Instrum.Meas., vol. 66, no. 3, pp. 535–542, Mar. 2017.

[12].Martinez-Hernandez,U.;Dehghani-Sanij,A.A. Adaptive Bayesian inference system for recognition of walking activities and prediction of gait events using wearable sensors. Neural Networks,2018,102,107-119.[13] Wahlström,J.;  Skog,I.;  Händel,P. Inertial Sensor Array Processing with Motion Models. In Proceedings of 2018 21st International Conference on Information Fusion (FUSION), Cambridge, UK, 2018, pp.788–793.

Point 2: Section of angular motion estimation method. Authors must add more discussions about the components of Figure 10.

Response 2: The solution flow of angular rate base on AUKF algorithm is shown in Fig.10.

Figure 10. The solution flow of angular rate base on AUKF algorithm.

First, the state value needs to be initialized to obtain the initial state vector and the initial error covariance matrix.Calculate the sigma point and weight value, and then calculate the one-step prediction status value.and are the result of propagation through the equation of state. Using the UT transform, the sigma sampling point is propagated through the measurement equation to obtain and, accelerometers and gyros provide measurements in the filtering algorithm and continuously update the measurement vector , and after obtaining the new measurement value, the filter is updated to obtain the filter gain value.From this, estimating the mean and covariance matrix of the process noise. Check if  is half positive definite, if not adjust the adaptive factor .Finally, the filtered state value and the covariance matrix are updated. The detailed derivation process of the algorithm is given in Section 5.2.

Point 3: Authors must include more discussions about the results of the Figure 14 and 21.

Response 3: Angular rate error at different rotating speed is shown in Figure.14.

Figure 14. Angular rate error at different rotating speed

It can be seen from Fig.14 that the estimation error of angular rate decreases as the rotational speed increases. When the rotation speed is less than 20r/s, the roll angular rate estimation error tends to be stable, and there is still a small oscillation characteristic due to the influence of process noise. As the rotational speed increases, the variation of process noise leads to the increase of estimation error of roll angle rate. As can be seen from the figure, the estimated error of the roll angular rate in one minute is less than 0.2 rad/s.When the rotational speed is greater than 25r/s, the angular rate estimation error shows a divergent trend. Therefore, the MEMS IMU device in this paper is more suitable for motion measurement of short-time working aircraft.

       (a)                                                                                         (b)

Figure 21. The Y-axis and z-axis angular rate and attitude angle in flight test: (a) The Y-axis and z-axis angular rate; (b) Yaw angle and Pitch angle.

 The Y-axis and z-axis angular rate and attitude angle in flight test are shown in Fig.21.The angular rate of the Y-axis and the Z-axis of the projectile is given in Fig.21 (a), and the attitude of the rocket projectile is calculated by the angular rate data. In the test, the ground frame is used as the navigation frame, and the test data is converted into the navigation frame by coordinate transformation, and then the attitude angle of the rocket projectile is calculated. From the change of the angular rate curve can be analyzed the rocket projectile’s flight status. Under the influence of the restraint of the launching cylinder, the angular motion of rocket projectile changes sharply in the first two seconds under the thrust of the solid engine. The maximum angular velocity reaches 60°/s, and the angular velocity tends to be stable after the exit, but subject to the external environment (such as wind speed, etc.), the rocket projectile begins to sway slightly during the flight.

It can be seen from Fig.21 (b) that the rocket projectile launch angle is 15°, which is the initial pitch angle. The attitude angle changes drastically within 2 seconds after launch. The pitch angle is slightly increased by the engine’s thrust, which indicating that the rocket projectile has a significant lift effect. After the engine’s thrust is over, the pitch angle gradually becomes smaller and the rocket projectile begins to fall. Under the influence of the restraint of the launching cylinder, the initial yaw angle of the rocket changes greatly. After the launch, the yaw angle gradually converges. Therefore, it is known that the pitch angle and yaw angle change after the end of the engine work is relatively stable, which indicating the basic stability characteristics of the rocket projectile during the flight test, it is beneficial to algorithm verification.

Point 4: Authors must mention the main limitations of their hollow structure MEMS-IMU device and AUKF algorithm.

Response 4: Thank you very much for your kind suggestion. The conclusion section of the manuscript supplements the main limitations of the hollow structure MEMS IMU device and AUKF algorithm. As follows

The hollow structure MEMS IMU device in my manuscript meets the installation requirements of the internal space of the rocket projectile.The MEMS IMU device with a hollow structure mainly uses the lever arm effect of the accelerometer to estimate the roll angular rate, to solve the problem that the roll angular rate of the projectile can’t be directly measured due to the gyro saturation. However, this hollow structure MEMS IMU also has certain limitations. Firstly, the low-cost MEMS sensors are adopted to ensure the calibration accuracy of the sensor. The IMU error model is complex and the calibration steps are complex. Secondly, installation error, coupling error and uncertainty error are difficult to eliminate, and the high-precision measurement of the MEMS IMU cannot be guaranteed. Finally, this kind of structure has poor universality and its application is limited by special environment. Therefore, it is necessary to select some high-precision MEMS sensors and research a high-precision calibration algorithm.

The traditional nonlinear filtering algorithm has the disadvantages of low precision and poor stability. In this paper, the AUKF algorithm has better adaptability to process noise estimation. The filtering accuracy is higher than the traditional nonlinear filtering algorithm and the stability is high. However, it also has certain limitations. It takes more time and memory to calculate the statistical error mean and covariance. Therefore, it is necessary to simplify the filtering model to reduce computation time and memory.

Once again, thank you very much for your comments and suggestions.

Best regards.

Reviewer 2 Report

The article is of overall good quality. However, I have some comments.

1.      The estimation algorithms presented in Section 4 are based on a least-squares method and could also be seen as computing maximum likelihood estimates under the assumption of Gaussian measurement noise. This approach to estimation using redundant accelerometer configurations has recently been applied in

J. Wahlström, I. Skog and P. Händel, "Inertial Sensor Array Processing with Motion Models," in 2018 21st International Conference on Information Fusion (FUSION), Cambridge, UK.

Please cite this article and clarify in what way the algorithms proposed in Section 4 differ from what is proposed in the above article. From what I can tell, it seems that you use the same approach but with a configuration that does not consist of three-axial sensors.

2.      Accelerometers measure specific force, not acceleration.

3.      I believe you are limiting yourself too much when talking about gyro-free IMUs in the introduction. It would be better to talk about “inertial arrays”, “redundant accelerometer configurations”, and “spatially distributed accelerometers”. The research on configurations consisting of spatially distributed accelerometers with gyroscopes typically go hand in hand with research on the same configurations without gyroscopes. The used terminology gets unnecessarily complicated when you talk about “gyro-assisted multi-accelerometer”, which is just an inertial array.

4.      Reg. “and thus a distributed accelerometers based inertial navigation system will have angular rate estimates that rapidly diverge, where the divergence rate is an order of magnitude greater than that of a gyro-equipped system“: This is a much too general statement. The latter part of the statement is only true when using less than nine accelerometers in the configuration. With nine accelerometers, it is possible to directly estimate angular rotations from spatially distributed accelerometers, and thereby avoid the extra integration step.

5.      The introduction should clarify that one of the main reasons to use redundant accelerometer configurations is to be able to estimate angular acceleration directly (without having to differentiate gyroscope measurements).

6.      Reg. “the accelerometer must be installed at the non-center of mass of the
projectile”: What is important is not the accelerometer’s positions with respect to the center of gravity, but their relative positions. The accelerometers must be spatially distributed (i.e., they cannot all have the same position) if we want to extract angular information from them.

7.      In Section 3, please specify how you have defined the error (is it rmse?) when specifying that it reduced by x percent.

8.      Please specify how you chose the angular rate reference curve shown in Figure 11.

9.      Use the established terminology “mean absolute deviation” and “root-mean-square error” in Table 2.

10.  The introduction would benefit from citing and drawing inspiration from the survey article

Nilsson, J.O.; Skog, I. Inertial sensor arrays - A literature review. In Proceedings of the 2016 European Navigation Conference (ENC), Helsinki, Finland, 30 May–2 June 2016; pp. 1–10.

Some comments on language:

1.      Page 1: The sentence “Three single-axis gyros are respectively mounted orthogonally to each other and are measure to the angular rates of the three orthogonal axes.” Does not seem to be grammatically correct. Please reformulate.

2.      Page 1: Reg. “an adaptive unscented kalman filter (AUKF) algorithm is proposed, which has higher estimation accuracy”: Please clarify what this algorithm has higher estimation accuracy than.

3.      Page 1: Specify that overload often also is called saturation.

4.      Page 2: MEME-IMU should be MEMS IMU.

5.      Kalman should always be written with a capital K.

Author Response

Point 1:  The estimation algorithms presented in Section 4 are based on a least-squares method and could also be seen as computing maximum likelihood estimates under the assumption of Gaussian measurement noise. This approach to estimation using redundant accelerometer configurations has recently been applied in

J. Wahlström, I. Skog and P. Händel, "Inertial Sensor Array Processing with Motion Models," in 2018 21st International Conference on Information Fusion (FUSION), Cambridge, UK.

Please cite this article and clarify in what way the algorithms proposed in Section 4 differ from what is proposed in the above article. From what I can tell, it seems that you use the same approach but with a configuration that does not consist of three-axial sensors.

Response 1: Thank you very much for your kind suggestion. The above article is cited in the following format:

[1] Wahlström,J.;  Skog,I.;  Händel,P. Inertial Sensor Array Processing with Motion Models. In Proceedings of 2018 21st International Conference on Information Fusion (FUSION), Cambridge, UK, 2018; pp.788–793.

In the above article, a maximum a posteriori (MAP) estimator for estimating the angular velocity, angular acceleration, and specific force of an inertial sensor array is presented. The maximum likelihood estimator is extended by introducing a motion model and deriving a maximum a posteriori estimator that jointly estimates the array dynamics at multiple sampling instances. Sensor geometry in the simulation is shown in Fig.1(a).

a. In the above article                            b. In my manuscript

Fig.1 Sensor geometry in the simulations

Comparing the configuration of the sensor, the difference between the two schemes is that the above document uses a gyro-free sensor array, in my manuscript, two gyro sensors are used to directly measure the yaw angle rate and pitch angle rate of the projectile, which is more accurate than the angular rate estimated by the accelerometer. The dual-axis accelerometers are only used to measure the roll angle rate. Comparing with the estimation accuracy of angular rate, the separation distance from the centroids is L=87.5px in my manuscript, the separation distance from the center of mass in the above article is 25px. According to the measurement model, the larger the distance, the higher the angular rate estimation accuracy, and the range of angular rate measurement is larger. In addition, two dual-axis accelerometers are adopted in my manuscript, comparing with the four accelerometers mentioned in the above article, which require less calculation and consume less power. Therefore, the sensor array distribution in my manuscript is more reasonable.

Point 2: Accelerometers measure specific force, not acceleration.

Response 2: Yes, you are right. The accelerometer does measure specific force. My description in this paper is wrong. Thank you for correcting it.

Point 3:  I believe you are limiting yourself too much when talking about gyro-free IMUs in the introduction. It would be better to talk about “inertial arrays”, “redundant accelerometer configurations”, and “spatially distributed accelerometers”. The research on configurations consisting of spatially distributed accelerometers with gyroscopes typically go hand in hand with research on the same configurations without gyroscopes. The used terminology gets unnecessarily complicated when you talk about “gyro-assisted multi-accelerometer”, which is just an inertial array.

Response 3: Thank you very much for your kind suggestions. For this problem, the discussion about “inertial arrays”, “redundant accelerometer configurations”, and “spatially distributed accelerometers” are supplemented in the introduction. Nine references are cited as follows:

[1] Wahlström,J.;  Skog,I.;  Händel,P. Inertial Sensor Array Processing with Motion Models. In Proceedings of 2018 21st International Conference on Information Fusion (FUSION), Cambridge, UK, 2018; pp.788–793.

[2]Nilsson,J.O.; Skog,I.  “Inertial sensor arrays — A literature review,” in European Navigation Conf., Helsinki, Finland, May 2016.

[3]Skog, I. J.; Nilsson,O.; Handel,P.; Nehorai,A.  Inertial sensor arrays, maximum likelihood, and Cramer-Rao bound,” ´ IEEE Trans. Signal Process., vol. 64, no. 16, pp. 4218–4227, Aug. 2016.

[4]Schwaab, M.; Reginya,S.A.; Sikora,A.; Abramov,E.V. “Measurement analysis of multiple MEMS sensor array,” in IEEE Int. Conf. Integr. Navigation Syst., Saint Petersburg, Russia, May 2017.

[5]Madgwick,S.O.H.; Harrison,A.J.L.; Sharkey,P.M.; Vaidyanathan,R.; Harwin,W.S. Measuring motion with kinematically redundant accelerometer arrays: theory simulation and implementation.Mechatronics.2013,23,518-529.

[6]Seyed Moosavi,S.;Moaveni,B.;Moshiri,B.;Arvan,M. Auto-Calibration and Fault Detection and Isolation of Skewed Redundant Accelerometers in Measurement While Drilling Systems. Sensors 2018,18,702.

[7] Jafari,M. Optimal redundant sensor configuration for accuracy increasing in space inertial navigation system. Aerosp. Sci. Technol. 2015, 47, 467–472.

[8] Kaswekar,P.; Wagner, J. F. Sensor fusion based vibration estimation using inertial sensors for a complex lightweight structure. IEEE 2015 DGON Inertial Sensors and Systems Symposium (ISS), Karlsruhe, Germany, 22-23 September, 2015.

[9]Brooks, I. M. Spatially distributed measurements of platform motion for the correction of ship-based turbulent fluxes, J. Atmos.Oceanic Technol, 2008, 25, 2007–2017.

About inertial arrays, literature [1-4] presents the current research status and related application problems. Madgwick [5] uses a redundant array of triple-axis accelerometers to measure the linear and angular motion of the rigid body. Seyed Moosavi [6] uses the redundant array of accelerometer to measure while drilling, multi-sensors fault detection and isolation are realized. The spatial distribution of the accelerometer is also discussed, which replaced the traditional orthogonal installation. Jafari [7] analyzes the performance of redundant IMUs and their various sensors configurations. The measurement accuracy can be improved by suitable geometric configuration. The structural vibration estimation method of Sofia airborne telescope is analysed [8], and the optimal configuration of spatially distributed accelerometers are used to estimate the vibration accurately. The motion correction of turbulence measurements on a mobile platform is realized by using a single-axis gyroscope and multiple spatially distributed accelerometers [9]. 

The description of “gyro-assisted multi-accelerometer” has been modified to “an inertial array”.

Point 4: Reg. ‘and thus a distributed accelerometers based inertial navigation system will have angular rate estimates that rapidly diverge, where the divergence rate is an order of magnitude greater than that of a gyro-equipped system’. This is a much too general statement. The latter part of the statement is only true when using less than nine accelerometers in the configuration. With nine accelerometers, it is possible to directly estimate angular rotations from spatially distributed accelerometers, and thereby avoid the extra integration step.

Response 4: Yes, that's true. There are some references to prove this. With nine accelerometers, it is possible to directly estimate angular rotations from spatially distributed accelerometers, and thereby avoid the extra integration step. Therefore, the above description  “ and thus a distributed accelerometers based inertial navigation system will have angular rate estimates that rapidly diverge, where the divergence rate is an order of magnitude greater than that of a gyro-equipped system” will be deleted in the revised manuscript.

Point 5: The introduction should clarify that one of the main reasons to use redundant accelerometer configurations is to be able to estimate angular acceleration directly (without having to differentiate gyroscope measurements).

Response 5: Yes, it should be described as such, one of the main reasons to use redundant accelerometer configurations is to be able to estimate angular acceleration directly, and the gyro is only an auxiliary measure. The corresponding modifications have been made in the revised manuscript.

Point 6: Reg. “the accelerometer must be installed at the non-center of mass of the projectile”: What is important is not the accelerometer’s positions with respect to the center of gravity, but their relative positions. The accelerometers must be spatially distributed (i.e., they cannot all have the same position) if we want to extract angular information from them.

Response 6: Yes, you are right. But, I just emphasize that the acceleration sensors should not be installed at the centroid position, which is not conducive to the sensitive specific force information for the accelerometer, thus affecting the solution accuracy of the angular velocity. In addition, relevant references have proved that the farther the installation point of the accelerometer is from the body's centroid position, the higher its measurement accuracy will be.

Point 7:  In Section 3, please specify how you have defined the error (is it rmse?) when specifying that it reduced by x percent.

Response 7: Take “Temperature compensation” example: as can be seen from Fig.5, the maximum of the acceleration output before temperature compensation is 1.8g, and the theoretical output is 0g. Therefore, defining the output error before compensation is 1.8g. The output error is compensated by the least squares method, and the maximum of the acceleration output after temperature compensation is 0.2g. Define the output error after compensation is 0.2g. According to the formula, the accelerometer error after compensation can be reduced by 89%.

Point 8: Please specify how you chose the angular rate reference curve shown in Figure 11.

Response 8: In general, computer simulation is used to verify the feasibility and accuracy of the algorithm in the environment of pure coning motion. In this paper, considering the motion characteristics of the high-speed spinning projectile, the spin angular rate is introduced, combined with the angular motion Eq. 62 under coning motion to draw the angular rate reference curve and further verify the accuracy of the angular rate based on AUKF algorithm.

Point 9: Use the established terminology “mean absolute deviation” and “root-mean-square error” in Table 2.

Response 9: Thank you very much for your correction. Table 2 is amended as follows:

Table2. Mean value and mean square deviation of angular rate estimated error

Rotation speed

r/s

Mean   value of error

Mean absolute deviation

(rad/s)

Mean square deviation

Root-mean-square error

()

5

0.0034

6.8120×10-6

10

0.0044

2.9833×10-5

15

0.0142

2.1568×10-4

20

0.0369

9.4818×10-4

25

0.0564

1.0352×10-3

30

0.0799

1.6753×10-3

Point 10: The introduction would benefit from citing and drawing inspiration from the survey article:

Nilsson, J.O.; Skog, I. Inertial sensor arrays - A literature review. In Proceedings of the 2016 European Navigation Conference (ENC), Helsinki, Finland, 30 May–2 June 2016; pp. 1–10.

Response 10:  Thank you very much for your kind suggestion. The above reference has been cited in the revised manuscript.

Some comments on language:

Point 1: Page 1: The sentence “Three single-axis gyros are respectively mounted orthogonally to each other and are measure to the angular rates of the three orthogonal axes.” Does not seem to be grammatically correct. Please reformulate.

Response 1: Modify the above sentence to:  Three single-axis gyros are mounted orthogonal to each other to measure the angular rate of the three axes respectively.”

Point 2:  Page 1: Reg. “an adaptive unscented kalman filter (AUKF) algorithm is proposed, which has higher estimation accuracy”: Please clarify what this algorithm has higher estimation accuracy than.

Response 2: Modify the above sentence to:  “which has higher estimation accuracy than UKF.”  It can be seen directly from Figure 13.

Point 3: Page 1: Specify that overload often also is called saturation

Response 3: Already modified.

Point 4: Page 2: MEME-IMU should be MEMS IMU

Response 4: Already modified.

Point 5: Kalman should always be written with a capital K

Response 5: Already modified.

Once again, thank you very much for your comments and suggestions.  I hope I can learn more from you.

Best regards.

Round  2

Reviewer 1 Report

This manuscript has been improved considering all the reviewer's comments. It is suitable to be published in Sensors.

Author Response

Dear reviewer:

        Thank you for your serious and constructive comments on our manuscript.

Our manuscript has been reviewed by a native English speaker and revised to improve readability. All changes have been highlighted in the revised manuscript.

Once again, thank you very much for your comments and suggestions.

Best regards.

Reviewer 2 Report

The authors have succesfully revised the paper according to the comments from the reviewer, and the paper is ready for publication.

I would give two minor comments:

1) Reg. your comment to point 6 in the authors' reply: It needs to be emphasized that the accelerometers positions with respect to each other matters for extracting the angular velocity. Therefore, their position with respect to the centroid point of the accelerometer positions matter. However, their positions with respect to the body's centroid position (defined as for example the center of mass) does not matter. Generally, the accuracy with which angular velocity can be estimated increases as the accelerometers are placed further from each other. However, the accuracy is not affected by the accelerometers placement with respect to the center of mass. 

2) The paper varies between presenting estimated angular veocity in rad/s and deg/s. You could think about being consistent in this respect. Especially fig. 14 may benefit from being shown in deg/s since the numbers in rad/s are very small. 

Author Response

Dear Reviewer:

Thank you for your serious and constructive comments on our manuscript.

Point 1: Reg. your comment to point 6 in the authors' reply: It needs to be emphasized that the accelerometers positions with respect to each other matters for extracting the angular velocity. Therefore, their position with respect to the centroid point of the accelerometer positions matter. However, their positions with respect to the body's centroid position (defined as for example the center of mass) does not matter. Generally, the accuracy with which angular velocity can be estimated increases as the accelerometers are placed further from each other. However, the accuracy is not affected by the accelerometers placement with respect to the center of mass.

Response 1: Thank you for your explanation. The description in the original manuscript has been rectified. We are grateful to the referees for pointing out this error.

Point 2:  The paper varies between presenting estimated angular veocity in rad/s and deg/s. You could think about being consistent in this respect. Especially fig. 14 may benefit from being shown in deg/s since the numbers in rad/s are very small.

Response 2: Thank you very much for your kind suggestion. All angular velocity units have been unified into deg/s in the updated manuscript.

Once again, thank you very much for your comments and suggestions.  

Best regards.